# Aβ initiates brain hypometabolism, network dysfunction and behavioral abnormalities via NOX2-induced oxidative stress in mice

Anton Malkov[1,6], Irina Popova[1,6], Anton Ivanov [2], Sung-Soo Jang[3], Seo Yeon Yoon[3], Alexander Osypov[1,4], Yadong Huang [3,5], Yuri Zilberter[2,6] & Misha Zilberter [3,6 ✉]

A predominant trigger and driver of sporadic Alzheimer's disease (AD) is the synergy of brain oxidative stress and glucose hypometabolism starting at early preclinical stages. Oxidative stress damages macromolecules, while glucose hypometabolism impairs cellular energy supply and antioxidant defense. However, the exact cause of AD-associated glucose hypometabolism and its network consequences have remained unknown. Here we report NADPH oxidase 2 (NOX2) activation as the main initiating mechanism behind $A\beta_{1-42}$-related glucose hypometabolism and network dysfunction. We utilize a combination of electrophysiology with real-time recordings of metabolic transients both ex- and in-vivo to show that $A\beta_{1-42}$ induces oxidative stress and acutely reduces cellular glucose consumption followed by long-lasting network hyperactivity and abnormalities in the animal behavioral profile. Critically, all of these pathological changes were prevented by the novel bioavailable NOX2 antagonist GSK2795039. Our data provide direct experimental evidence for causes and consequences of AD-related brain glucose hypometabolism, and suggest that targeting NOX2-mediated oxidative stress is a promising approach to both the prevention and treatment of AD.

[1] Institute of Theoretical and Experimental Biophysics, Russian Academy of Sciences, Pushchino, Russia. [2] Aix Marseille Université, Inserm, Marseille, France. [3] Gladstone Institute of Neurological Disease, San Francisco, CA, USA. [4] Institute of Higher Nervous Activity and Neurophysiology, Russian Academy of Sciences, Moscow, Russia. [5] Department of Neurology, University of California, San Francisco, CA, USA. [6] These authors contributed equally: Anton Malkov, Irina Popova, Yuri Zilberter, Misha Zilberter. ✉email: misha.zilberter@gladstone.ucsf.edu

The pathological processes that drive AD may begin several decades before the first clinical symptoms manifest[1–3]. Meanwhile, numerous human studies suggest a causal upstream role for Aβ in AD initiation[3–5], and recent progress in AD biomarkers has revealed that β-amyloidosis is one of the earliest signs of AD pathogenesis, detectable at preclinical stages of the disease[1–3]. Interestingly, at this stage patients are still cognitively unimpaired but may demonstrate variable neuropsychiatric symptoms (e.g., apathy, depression, agitation/aggression, anxiety, and irritability)[6,7], the emergence of which may correlate with amyloidosis[8]. Although Aβ toxicity alone may not be sufficient to cause cognitive deterioration, it likely triggers downstream pathological changes (i.e., tauopathy and neurodegeneration) that ultimately lead to cognitive decline at later stages[3,9].

Early Aβ pathology is often detected concurrently with glucose hypometabolism[10–13], another pre-symptomatic marker of AD. Glucose hypometabolism is implicated in the initiation of sporadic AD as it is associated with most major AD risk factors[10–13]. It occurs in patients with amnestic mild cognitive impairment (aMCI), widely thought to be a prodromal stage of AD[10,14,15], and has also been detected in AD patients almost two decades prior to the onset of clinical symptoms. Furthermore, glucose hypometabolism can be an accurate predictive marker for AD development in aMCI patients[10,14,15]. Since glucose utilization underlies vital brain functions such as energy supply and antioxidant defense[16], it is not surprising that disturbances in glucose metabolism can lead to a chain of harmful consequences, and thus likely represent a major underlying cause of disease initiation and progression[12]. However, until now the exact causes and consequences of AD-associated glucose hypometabolism have been unknown, hampering the search for effective treatment.

Multiple studies have shown that oligomeric Aβ induces brain oxidative stress[12,17,18] largely via activation of NADPH oxidase (NOX)[19–21] in different cell types. NOX is the only known enzyme with the primary function of generating ROS[22]. NOX family of enzymes are transmembrane carriers that transport an electron from cytosolic NADPH to reduce oxygen to superoxide anion. There are seven isoforms of NOX with NOX1, NOX2 and NOX4 expressed in multiple brain regions including the cerebral cortex, hippocampus, cerebellum, hypothalamus, midbrain, and/or striatum[23], with NOX2 the dominant form expressed by microglia, neurons, and astrocytes[23–25]. Several lines of evidence including post-mortem analyses of AD patients' cerebral cortices indicate that oxidative stress—particularly resulting from NOX2 activation—plays a significant role in the development of AD[20,23,25–27]. Close relationship between the levels of Aβ and NOX2 activity has also been well documented in multiple studies[19,28–33].

We now show that NOX activation by oligomeric Aβ₁₋₄₂ results in pathological changes in brain glucose consumption, hippocampal network hyperactivity, and neuropsychiatric-like disturbances in mouse behavior. All observed abnormalities were prevented by blocking NOX2, suggesting NOX may be a major molecular mechanism behind AD initiation and progression. Our results point to early intervention in NOX-induced oxidative stress as a potential effective approach to AD prevention and treatment.

## Results

### Aβ₁₋₄₂ disrupts network glucose utilization both in brain slices and in vivo.
Fibrillar Aβ₁₋₄₂ (400 nM) applied to hippocampal slices reduced network activity-driven glucose uptake nearly by half (Fig. 1A and Table 1.1), an effect paralleled by and correlated

with decreased release of lactate (Table 1.2 and Supplementary Fig. 1A–C). To confirm our results in vivo, we injected Aβ₁₋₄₂ i.c.v. in anesthetized mice; following Aβ₁₋₄₂ injection, extracellular glucose transients in response to synaptic stimulation changed more than twofold (Fig. 1B and Table 1.3), indicating a reduction in glucose uptake of similar magnitude to what we observed in brain slices. In the living brain, extracellular glucose is the product of interplay between glucose supply from blood and cellular glucose uptake. Network activation leads to an increase in glucose uptake but the blood vessels rapidly dilate and the supply of glucose is upregulated to compensate, leading to an actual net increase of extracellular glucose as has been demonstrated previously[34,35]. Since Aβ did not affect glucose delivery—network local field potential (LFP) response to stimulation did not change significantly (Fig. 1Bv, Table 1.4), nor did the baseline glucose concentration (see for example Fig. 1Bii)—a doubling in extracellular glucose transient amplitude indicates reduced glucose uptake. Aβ₁₋₄₂-induced glucose hypometabolism was underlain by decreased glycolysis, seen as reduced amplitudes of NAD(P)H fluorescence overshoot (Fig. 1C and Table 1.5) together with an increase of oxidation (dip) amplitude[36] (Fig. 1C and Table 1.6). This was paralleled by a decrease in FAD fluorescence undershoot phase (Table 1.7 and Supplementary Fig. 1F), further confirming disrupted glycolysis[36]. Aβ₁₋₄₂ also induced a significant increase in activity-driven oxygen consumption (Fig. 1D and Table 1.8) that positively correlated with changes in NAD(P)H oxidation phase (Supplementary Fig. 1D), an association that likely indicates upregulated mitochondrial respiration to compensate for reduced glycolysis[37,38].

Meanwhile, none of the above disruptions could be attributed to changes in network activity, as the stimulus train LFP integral did not change (Fig. 1E and Table 1.9), nor was there any significant correlation between observed metabolic changes and changes in network response (Supplementary Fig. 1E).

To test whether glucose hypometabolism could instead be due to reduced glucose transport as suggested previously[39–41], we repeated NAD(P)H and FAD fluorescence recordings after increasing ACSF glucose concentration from 5 mM to 10 mM; if Aβ exerts its effect by inhibiting the passive glucose flux, then increasing extracellular glucose would serve to at least partially normalize that effect. However, doubling the glucose concentration in ACSF had no restorative effect on neither the disrupted NADH(P)H nor FAD transients (Supplementary Fig. 1F), suggesting Aβ-induced glucose uptake reduction is indeed underlain by reduced glycolysis and not by impaired glucose transport.

### Aβ₁₋₄₂ toxicity is mediated by NOX2.
Aβ₁₋₄₂-induced changes in glucose utilization were strikingly similar to those we previously observed following seizures that were preceded by a spike in extracellular H₂O₂[42]. In that context, the likely source of the H₂O₂ and actual trigger of seizures is the reactive oxygen species (ROS)-generating NOX, specifically NOX2[43]. Aβ₁₋₄₂ has been reported to induce oxidative stress in multiple cell types by activating NOX[18–21]. In our hands, when NOX2 was inhibited by the selective antagonist GSK2795039, Aβ₁₋₄₂ failed to disrupt glucose utilization in slices (Fig. 2A and Table 1.10) or in vivo (Fig. 2B and Table 1.11). NOX2 inhibition also prevented Aβ₁₋₄₂-induced modification of glycolysis (Fig. 2C and Table 1.13–14) and oxygen consumption (Fig. 2D and Table 1.15). Meanwhile, network activity response did not change significantly following GSK2795039 or subsequent Aβ₁₋₄₂ application (Fig. 2E and Table 1.16), suggesting that changes in network response could not be responsible for the normalizing GSK2795039 effect. Finally, we repeated our experiments in slices from NOX2-

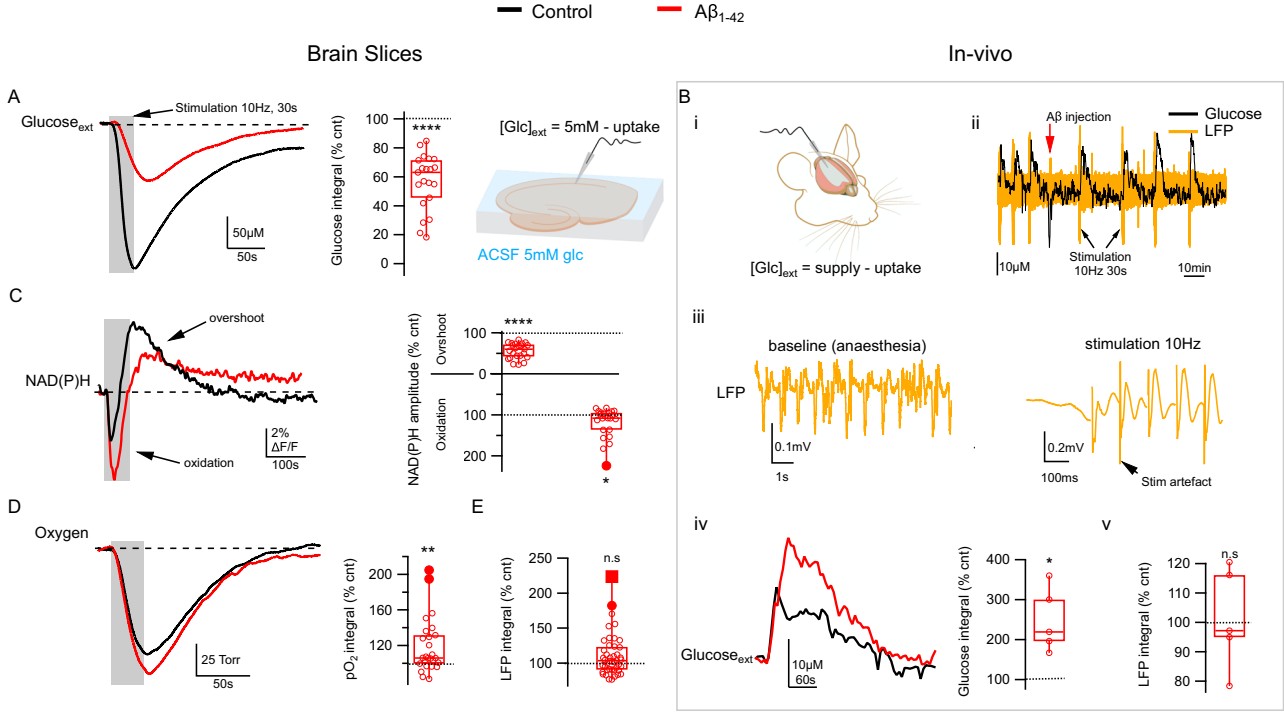

**Fig. 1 Aβ$_{1-42}$ inhibits network glucose utilization. A** In brain slices, 40 min of fibrillar Aβ$_{1-42}$ application reduces network activity-driven glucose uptake. Left, example traces from a single experiment showing the extracellular glucose transients in CA1 pyramidal cell layer in response to a 10 Hz, 30 s stimulation of Schaffer collaterals (gray) in control (black) and following 40 min of Aβ$_{1-42}$ application (red). Considering the constant 5 mM glucose supply from the perfusate, the drop in the transient amplitude indicates reduced uptake. Middle, summary graph showing glucose transient integral values normalized to controls. **B** In anesthetized mice, i.c.v. injection of fibrillar Aβ$_{1-42}$ results in a rapid change in glucose uptake. (i) schematic depicting the relationship between the dynamic extracellular glucose supply from the blood (which increases following synaptic activation) and network glucose uptake. (ii) example traces from a single experiment showing both local field potential (LFP, orange) and extracellular glucose (black) recordings from the CA1 region before and after Aβ$_{1-42}$ injection. (iii) detail of LFP recording showing characteristic anesthesia-induced oscillations (left) at baseline and the response to synaptic stimulation (right). (iv) average stimulation-induced glucose transients from a single experiment in control (black) and following i.c.v. Aβ$_{1-42}$ injection (red) and summary graph showing glucose transient integral values normalized to controls for all experiments; (v) summary of LFP integrals showing that stimulation response did not change significantly following Aβ$_{1-42}$ injection; this suggests that the activity-induced increase in glucose supply from the blood did not change, and therefore the apparent increase in glucose transients following Aβ$_{1-42}$ injection indicates reduced uptake. **C** Aβ$_{1-42}$ inhibits activity-driven glycolysis. Left, NAD(P)H autofluorescence traces from a single experiment in control (black) and following 40 min of Aβ$_{1-42}$ application (red). Right, summary values of transient amplitudes both for the overshoot (glycolysis-related) and oxidation phases of the signal. **D** Aβ$_{1-42}$ increases oxygen consumption: sample pO$_2$ traces from a single experiment showing a transient decrease of tissue oxygen levels in control (black) and following 40 min of Aβ$_{1-42}$ application (red) and a summary plot of normalized pO$_2$ integral values. **E** Aβ$_{1-42}$ does not significantly change the network response to synaptic stimulation: a summary plot of normalized LFP train integral values. Data are presented as box plots with min-max whiskers and Tukey quartile method. *$p < 0.05$, **$p < 0.01$, ***$p < 0.001$, ****$p < 0.0001$.

deficient (Cybb$^{tm1din}$/J) mice. Aβ$_{1-42}$ failed to exert any detrimental effect in any of the observed parameters (Fig. 2F and Table 1.17–21), confirming that Aβ$_{1-42}$ inhibits glucose metabolism specifically via NOX2 activation-induced oxidative stress.

We further examined the role of NOX2 in oxidative stress induced by Aβ$_{1-42}$ through additional biochemical analysis. Brain slices exposed to Aβ$_{1-42}$ exhibited augmented oxidative stress seen as higher levels of malondialdehyde (MDA), a by-product of lipid peroxidation and a standard oxidative stress marker (Table 1.22 and Supplementary Fig. 2). The Aβ$_{1-42}$-induced increase in MDA levels was prevented by NOX2 inhibition by GSK2795039 (Table 1.23 and Supplementary Fig. 2) and was also absent in Cybb$^{tm1din}$/J mouse slices (Table 1.24 and Supplementary Fig. 2), confirming that activated NOX2 is the primary source of overall acute oxidative stress induced by Aβ$_{1-42}$.

We also tested whether alternative pharmacological activation of NOX2 would mimic the effect of Aβ$_{1-42}$ on glucose utilization. Standard NOX activator PMA (100 nM[44]) reproduced the Aβ$_{1-42}$ effects in brain slices, resulting in the reduction of both glucose

consumption (Supplementary Fig. 3A, C) and NAD(P)H overshoot (Supplementary Fig. 3B, C). Accordingly, NOX2 antagonist GSK2795039 completely prevented the effects of PMA (Supplementary Fig. 3B–F).

In the brain, NOX2 is largely expressed in microglia, the resident brain phagocytes, where it is used as a host defense mechanism[26,27]. However, our experiments suggest that neurons[45] play the dominant role in generating Aβ$_{1-42}$/NOX2-induced ROS toxicity. While NOX2 in microglia is activated in response to neurotoxic stimulation[25], NMDA receptor stimulation is required for NOX2 activation in neurons[46], and we found that NMDAR blockade recapitulated the effect of NOX2 inhibition (Supplementary Fig. 4). Additionally, depleting microglia through dietary PLX5622 supplementation did not ameliorate the Aβ$_{1-42}$ effect on glycolysis (as NAD(P)H transients; Supplementary Fig. 4B) which was again fully prevented by GSK2795039 (Supplementary Fig. 4B), indicating a significant presence of functional NOX2 in our microglia-depleted slices. However, we cannot rule out a possible contribution of microglia

**Table 1 Summary of all recorded parameters, presented as percentage change from controls.**

| | Parameter | Change from control, % | SEM | n | p value | Fig. |
|---|---|---|---|---|---|---|
| 1 | Extracellular glucose transient amplitude, slices, Aβ | 57.7 | 4.13 | 21 | <0.0001 | 1A |
| 2 | Extracellular lactate transient amplitude, slices, Aβ | 67 | 9.85 | 7 | <0.02 | S1A,B |
| 3 | Extracellular glucose transient amplitude, in vivo, Aβ | 248.5 | 45 | 5 | <0.0005 | 1Biv |
| 4 | LFP train integral, in vivo, Aβ | 101.4 | 7.644 | 5 | 0.8597 | 1Bv |
| 5 | NAD(P)H overshoot amplitude, Aβ | 56.4 | 3.8 | 23 | <0.0001 | 1C |
| 6 | NAD(P)H dip amplitude, Aβ | 118.6 | 7.55 | 23 | <0.03 | 1C |
| 7 | FAD undershoot amplitude, Aβ | 73.1 | 4.32 | 7 | <0.001 | S1F |
| 8 | pO2 integral, Aβ | 118.7 | 6 | 26 | <0.005 | 1D |
| 9 | LFP train integral, slices, Aβ | 102.6 | 3.96 | 21 | 0.11 | 1E |
| 10 | Extracellular glucose transient amplitude, slices, GSK2795039 + Aβ | 109.5 | 10.2 | 7 | 0.39 | 2A |
| 11 | Extracellular glucose transient amplitude, in vivo, GSK2795039 + Aβ | 91.44 | 5.94 | 5 | 0.22 | 2Biii |
| 12 | LFP train integral, in vivo, GSK2795039 + Aβ | 126.8 | 27.3 | 5 | 0.625 | 2Biv |
| 13 | NAD(P)H overshoot amplitude, GSK2795039 + Aβ | 125.2 | 16.47 | 8 | 0.17 | 2C |
| 14 | NAD(P)H dip amplitude, GSK2795039 + Aβ | 103.6 | 5.841 | 8 | 0.5598 | 2C |
| 15 | pO2 integral, GSK2795039 + Aβ | 100 | 3.677 | 4 | 0.995 | 2D |
| 16 | LFP train integral, slices, GSK2795039 + Aβ | 82.09 | 4.711 | 8 | <0.001 | 2E |
| 17 | Extracellular glucose transient amplitude, Aβ in Cybb(tm1din/J) mice | 100.6 | 6.41 | 6 | 0.9309 | 2Fi |
| 18 | NAD(P)H overshoot amplitude, Aβ in Cybb(tm1din/J) mice | 93.15 | 9.079 | 10 | >0.9999 | 2Fii |
| 19 | NAD(P)H dip amplitude, Aβ in Cybb(tm1din/J) mice | 107 | 4.527 | 10 | 0.1572 | 2Fii |
| 20 | pO2 integral, Aβ in Cybb(tm1din/J) mice | 110.8 | 8.525 | 10 | 0.239 | 2Fiii |
| 21 | LFP train integral, Aβ in Cybb(tm1din/J) mice | 114 | 7.044 | 18 | 0.0639 | 2Fiii |
| 22 | MDA, Aβ | 273.2 | 101.4 | 7 | <0.05 | S2 |
| 23 | MDA, GSK2795039 + Aβ | 72.51 | 8.39 | 6 | <0.03 | S2 |
| 24 | MDA, Aβ in Cybb(1tmDin/J) mouse slices | 107 | 18.57 | 6 | 0.7201 | S2 |
| 25 | Accumulated Activity, 1 h post-injection, i.c.v. Aβ | 325 | 60.79 | 14 | <0.003 | 3C |
| 26 | Interictal spike rate, 1 h post-injection, i.c.v. Aβ | 674.2 | 172.1 | 16 | <0.002 | 3C |
| 27 | Accumulated Activity, 48 h post-injection, i.c.v. Aβ | 184.3 | 21.55 | 14 | <0.002 | 3D |
| 28 | Interictal spike rate, 48 h post-injection, i.c.v. Aβ | 179.9 | 35.04 | 14 | <0.05 | 3D |
| 29 | pHFO rate, 48 h post-injection, i.c.v. Aβ | 304.4 | 66.26 | 13 | <0.01 | 3D |
| 30 | Accumulated Activity, 1 h post-injection, i.c.v. GSK2795039 + Aβ | 134.4 | 41.98 | 9 | 0.4357 | 3C |
| 31 | Interictal spike rate, 1 h post-injection, i.c.v. GSK2795039 + Aβ | 269.2 | 73.05 | 10 | 0.0977 | 3C |
| 32 | Accumulated Activity, 48 h post-injection, i.c.v. GSK2795039 + Aβ | 47.53 | 12.07 | 8 | <0.004 | 3D |
| 33 | Interictal spike rate, 48 h post-injection, i.c.v. GSK2795039 + Aβ | 11.16 | 3.412 | 8 | <0.008 | 3D |
| 34 | pHFO rate, 48 h post-injection, i.c.v. GSK2795039 + Aβ | 88.18 | 28.23 | 10 | 0.6853 | 3D |
| 35 | Accumulated Activity, 1 h post-injection, i.c.v. Vehicle | 114.4 | 27.19 | 10 | 0.6092 | 3C |
| 36 | Interictal spike rate, 1 h post-injection, i.c.v. Vehicle | 105.6 | 14.74 | 12 | 0.5557 | 3C |
| 37 | Accumulated Activity, 48 h post-injection, i.c.v. Vehicle | 107 | 33.97 | 10 | 0.8422 | 3D |
| 38 | Interictal spike rate, 48 h post-injection, i.c.v. Vehicle | 107.3 | 29.06 | 11 | 0.9502 | 3D |
| 39 | pHFO rate, 48 h post-injection, i.c.v. Vehicle | 43.65 | 13.3 | 7 | <0.006 | 3D |

as they have also been reported to express NMDARs[47] (although the presence of functional microglial NMDARs in situ is a matter of debate[48,49]). Moreover, our microglial depletion experiments did not achieve complete ablation (Supplementary Fig. 4B) and with the Iba1 staining the potential presence of infiltrating macrophages can not be excluded.

**NOX2 inhibition prevents Aβ$_{1-42}$-induced hyperexcitability in vivo.** Epileptic seizures are a frequent comorbidity of AD[50,51]. Epilepsy also occurs in multiple transgenic mouse models over-producing Aβ[52], and network hyperactivity has been observed in hippocampal slices following acute Aβ application[53,54]. To investigate the immediate and medium-term effects of Aβ$_{1-42}$ on network electrophysiology, we recorded hippocampal CA1 local field potentials in awake freely moving mice before and following i.c.v. oligomeric Aβ$_{1-42}$ injection. We observed a rapid onset of network hyperactivity following Aβ$_{1-42}$ injection, seen both as an increase in accumulated activity (a general measure of all network activity exceeding 3xSD[55]) and in interictal spike rate (Fig. 3A, C and Table 1.25 and 26). Network hyperactivity persisted at least up to 48 h (Fig. 3Di, ii and Table 1.27 and 28) after the injection. Furthermore, Aβ$_{1-42}$ induced a significant increase in CA1 pathological high-frequency oscillations (pHFOs; 250-600 Hz), a marker for epileptogenesis[56,57] (Fig. 3Diii and Table 1.29).

Importantly, NOX2 inhibition by i.c.v application of GSK2795039 both prior to and following Aβ$_{1-42}$ injection completely prevented all network abnormalities, both acute and 48 h later (Fig. 3B–D and Table 1.30–34). This suggests that NOX2 activation is crucial not only for Aβ$_{1-42}$-induced brain glucose hypometabolism but also for Aβ$_{1-42}$-induced network abnormalities, highlighting the causal link between the two pathologies. It is also in line with our previous reports showing that NOX2 is the primary trigger of epileptic seizures[43] and that induced brain glucose hypometabolism results in network hyperactivity and epileptogenesis[35].

**NOX2 inhibition prevents behavioral and psychological symptoms induced by Aβ.** We noticed that Aβ-injected mice exhibited clearly increased aggression and anxiety, and evaluated these mouse behaviors 1 week (Open Field and Social Interaction tests) and 2 weeks (Partition Test) after i.c.v. administration of Aβ$_{1-42}$ (Fig. 4A). In the Open Field Test[58], Aβ-injected mice spent twice as long in the central part of the field compared to control mice, signifying elevated anxiety that was further demonstrated by increased defecation (Fig. 4B). Mice in the Aβ-group also displayed a much higher level of aggression as revealed by the Social Interaction-Fights test (see Supplementary Movie 1), with the proportion of aggressors as well as the number and total duration of fights increasing sharply (Fig. 4Ci). This was

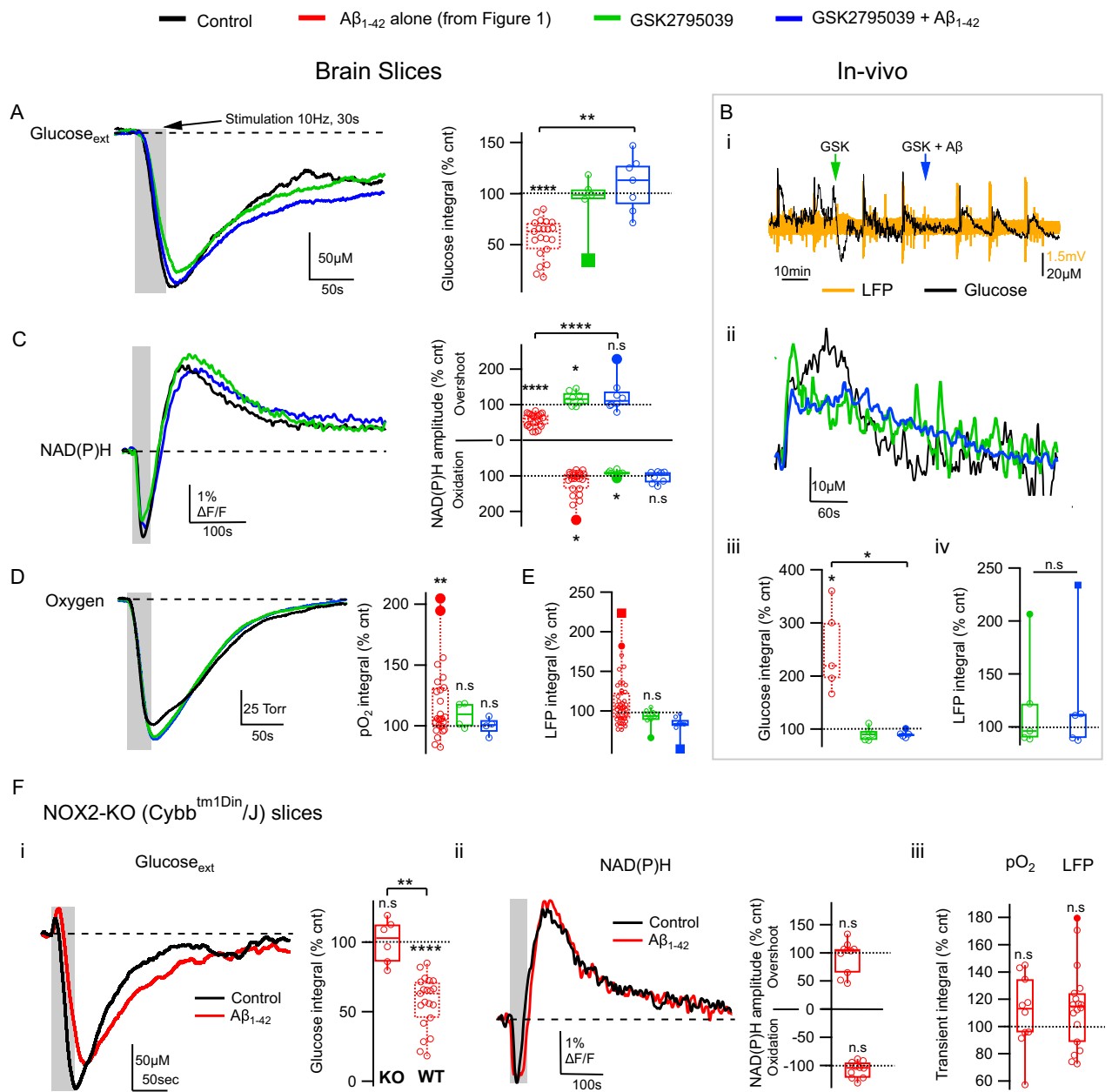

**Fig. 2 NOX2 is critical for Aβ$_{1-42}$–induced disruption of network glucose utilization. A** In brain slices, application of NOX2 antagonist GSK2795039 prevents the reduction in network activity-driven glucose uptake caused by Aβ$_{1-42}$: example traces from a single experiment showing the extracellular glucose transients in CA1 pyramidal cell layer in response to a 10 Hz, 30 s stimulation of Schaffer collaterals (gray) in control (black), after wash-in of GSK2795039 (green), and following 40 min of GSK2795039 + Aβ$_{1-42}$ application (blue); summary graph showing corresponding glucose transient integral values normalized to controls with those of Aβ$_{1-42}$ alone for comparison (red). **B** In anesthetized mice, intraventricular injection of GSK2795039 + Aβ$_{1-42}$ (blue) preceded by GSK2795039 (green) fails to elicit any changes in glucose uptake. (i) example traces from a single experiment showing both local field potential (LFP, orange) and extracellular glucose (black) recordings from the CA1 region. (ii) average stimulation-induced glucose transients from a single experiment in control (black), following i.c.v. GSK2795039 injection (green) and subsequent injection of GSK2795039 + Aβ$_{1-42}$ (blue). (iii) the summary graph showing corresponding glucose transient integral values together with those of Aβ$_{1-42}$ alone for comparison (red). (iv) summary of LFP integral values showing no significant change following either injection. **C** GSK2795039 prevents Aβ$_{1-42}$ disruption of activity-driven glycolysis. Left, NAD(P)H autofluorescence traces from a single experiment in control (black) after wash-in of GSK2795039 (green), and following 40 min of GSK2795039 + Aβ$_{1-42}$ application (blue); Right, summary values of transient amplitudes both for the overshoot (glycolysis-related) and "oxidation" phases of the signal. **D** GSK2795039 blocks Aβ$_{1-42}$ induced increase in oxygen consumption: sample pO$_2$ traces from a single experiment showing a transient decrease of tissue oxygen levels in control (black) after wash-in of GSK2795039 (green), and following 40 min of GSK2795039 + Aβ$_{1-42}$ application (blue); the summary plot of normalized pO$_2$ integral values. **E** GSK2795039 does not significantly modify network response to stimulation (green), while under Aβ + GSK2795039, the response decreases (blue); summary plot of stimulation train LFP integral values with Aβ-only values for comparison (dotted red). **F** Aβ$_{1-42}$ application has no effect in NOX2-deficient mouse slices: sample traces and summary of glucose (i) and NAD(P)H fluorescence transients (ii), as well as summary of changes in oxygen consumption and LFP integrals (iii). Data are presented as box plots with min-max whiskers and Tukey quartile method. *$p < 0.05$, **$p < 0.01$, ***$p < 0.001$, ****$p < 0.0001$.

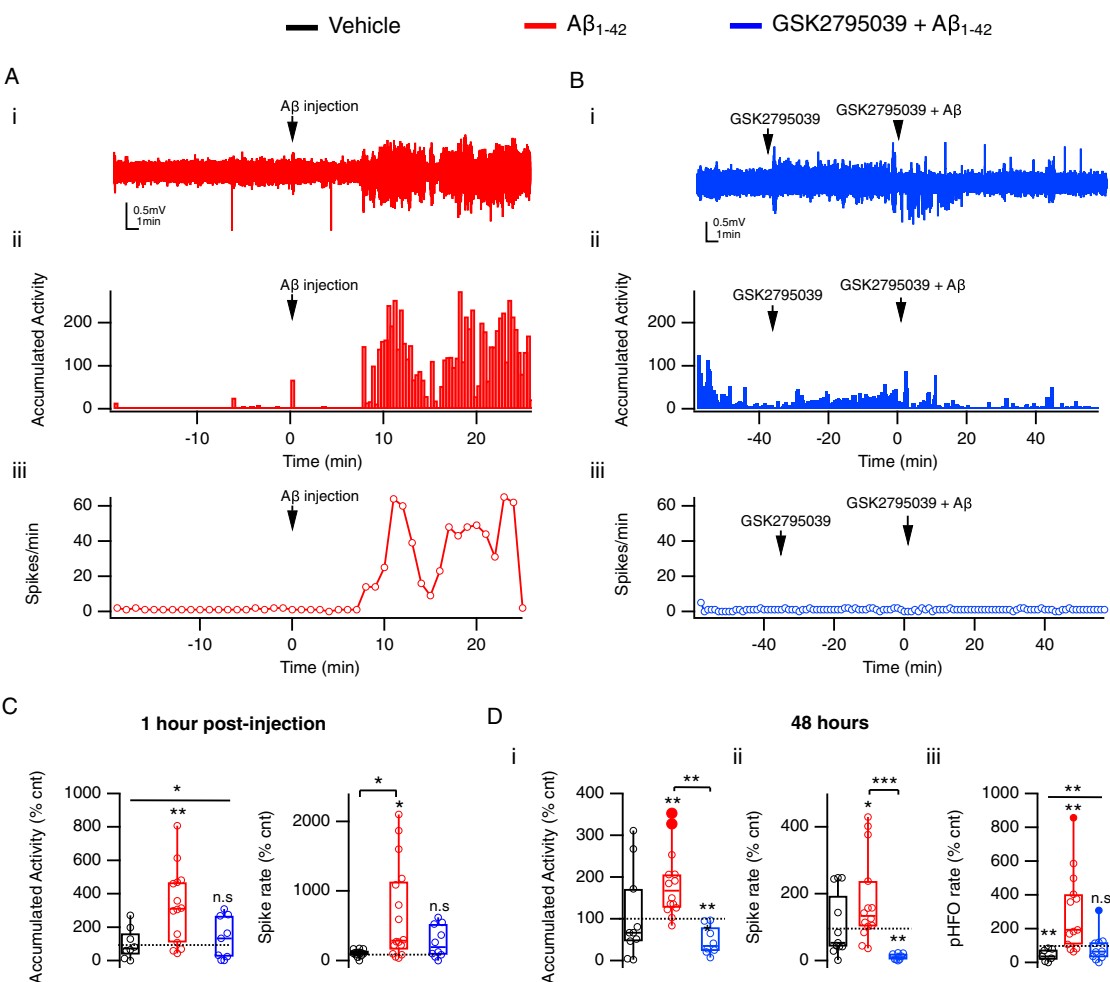

**Fig. 3 NOX2 mediates Aβ effect on network activity in vivo. A** Aβ$_{1-42}$ i.c.v. injection results in an acute increase in network activity and interictal spike frequency in awake freely moving mice. (i) Example LFP trace from hippocampal CA1 stratum pyramidale before and after Aβ injection. (ii) 20-s accumulated activity integrals analyzed from the top trace. (iii) Interictal spike frequency analyzed from the top trace. **B** NOX2 blockade prevents Aβ-induced hyperactivity. (i) Example LFP trace from hippocampal CA1 stratum pyramidale in control, following GSK2795039 injection, and after subsequent Aβ injection. (ii) Accumulated activity integrals analyzed from the top trace. (iii) Interictal spike frequency analyzed from the top trace. **C** Mean 1-h accumulated activity integral and interictal spike rate values for all experiments in **A**, **B**. **D** Aβ-induced hyperactivity persists 48 h following the Aβ injection and is prevented by preceding GSK2795039 application. (i) Average accumulated activity integrals. (ii) Average interictal spike rates. (iii) Average pHFO rates. Data are presented as box plots with min-max whiskers and Tukey quartile method. *$p < 0.05$, **$p < 0.01$, ***$p < 0.001$.

paralleled by reduced positive social interactions in Aβ-group: while vehicle-injected mice tended to spend a significant proportion of their time huddled in groups, Aβ-injected mice spent almost no time "cuddling" (Fig. 4Cii and Supplementary Movie 1). Increased aggression in Aβ-injected mice was also confirmed by the modified Partition Test[59], where they spent a significantly longer time in direct proximity/contact with the partition separating them from the reference mouse in effort to reach it (Fig. 4D). Altogether, these tests demonstrated anxiety and marked aggression in male mice following Aβ$_{1-42}$ injection (see also refs. [60,61]). Importantly, all observed behavioral abnormalities were prevented by daily i.c.v. injections of GSK2795039 (Fig. 4B–D), indicating that these neuropsychiatric-like disturbances were caused by NOX2-induced oxidative stress.

## Discussion
Aβ accumulation and glucose hypometabolism are early AD biomarkers implicated in disease initiation, however the exact causal relationship between the two remained unclear. Multiple clinical studies in MCI and early AD patients demonstrated a negative correlation between regional glucose metabolism and amyloid load[10,62,63]. Glucose hypometabolism possesses a specific distribution pattern in the brain including precuneus and posterior cingulate cortex, extending to occipital lobes, medial and lateral frontal lobes, and middle temporal gyrus, and is consistent with the pattern of amyloid deposition[62,63]. In addition, combined FDG-PET and CSF Aβ$_{1-42}$ biomarkers have been shown to be predictive of the progression risk to AD in MCI subjects[10,14,15]. Meanwhile, at later AD stages some studies reported poor regional correlation between hypometabolism and amyloid load[62,64–66], although global levels remained correlated[10,67–70]. There may be several reasons for such discrepancy, the most likely being amyloid presence in the cortex long before the metabolic measurements, but also remote effects of amyloid deposition[71] and potential differences in detection thresholds of imaging modalities, etc.[62]

A parallel early AD co-morbidity is acquired epilepsy[72,73] underlain at least in part by Aβ-induced network hyperexcitability[53,54,74]. Interestingly, most acquired epilepsy patients outside of AD also exhibit interictal brain glucose hypometabolism that can be predictive of disease initiation (for review,

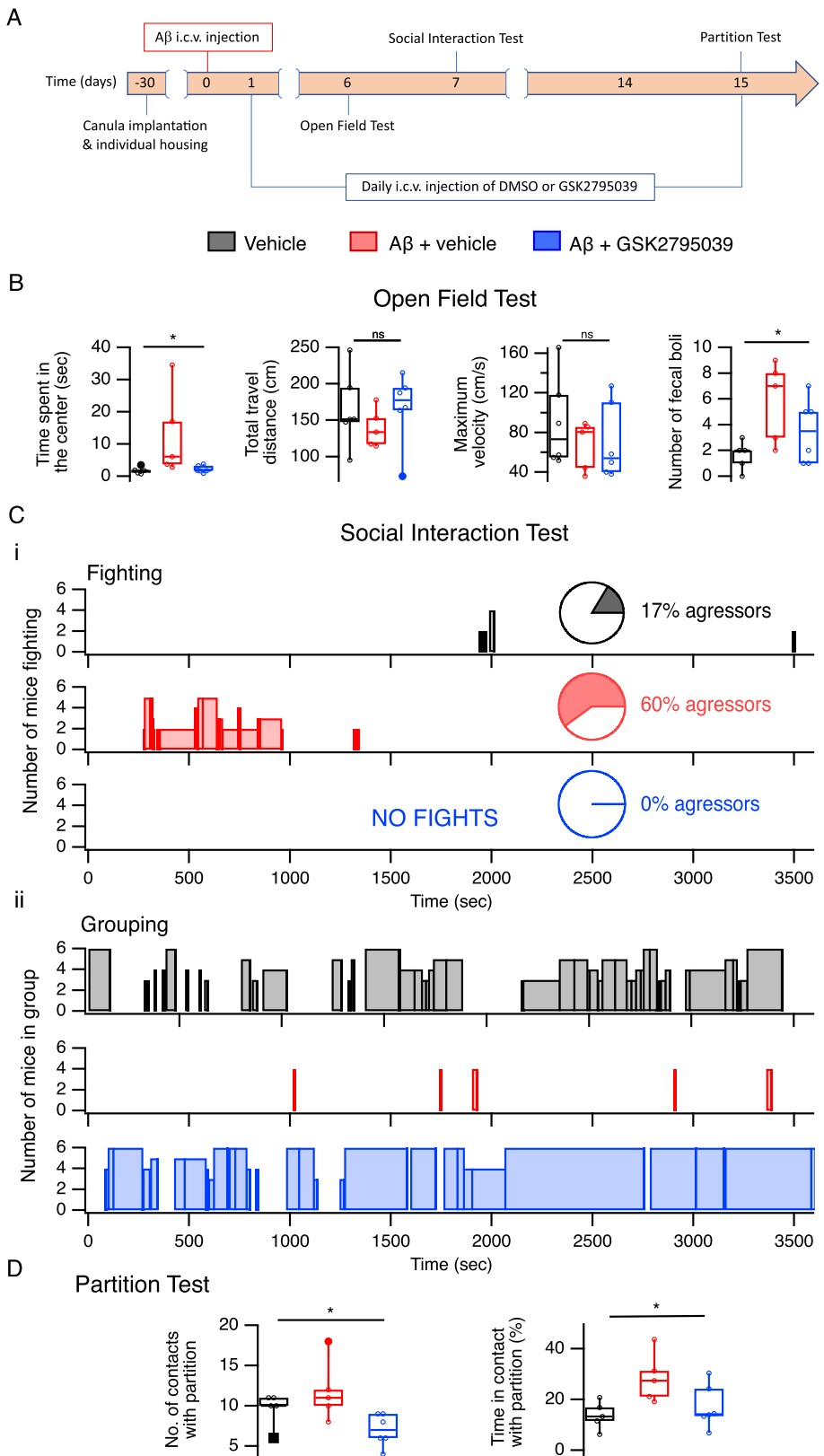

ref. [75]). We reported previously that Aβ-induced network hyper-excitability is paralleled by disruptions in glycolysis[53] and that normalizing this reduction in glycolysis by chronic exogenous energy substrate (pyruvate) administration abolished the epileptic phenotype in AD model mice[53,55]. Outside of Aβ toxicity, we have shown that chronic artificial inhibition of brain glycolysis by i.c.v.

2-deoxyglucose (2-DG) injections in healthy rats resulted in epileptogenesis and seizures (Samokhina et al.[35], see also Samokhina et al.[76]), suggesting that brain glucose hypometabolism, no matter its triggers, results in network hyperactivity.

Importantly, brain hypometabolism in AD is always associated with oxidative stress (for review, refs. [12,15]). Energy deficiency[77,78]

**Fig. 4 NOX2 blockade prevents Aβ[1-42]–induced neuropsychiatric-like behavioral abnormalities. A** Timeline diagram of experimental design. **B** Open field test. Animals were placed in the center of the OF and latent time (sec) of the first run from the central square of 20 × 20 cm was recorded. Such indicators of anxiety as the number of entries into the center (n), the average time in the center (s), and the number of fecal boli (n) are also presented. **C** Social Interaction test: animals, individually housed for 5 days prior, were placed on a round field for 1 h. (i) Fighting Test**:** the number of fights, the number of aggressors, and duration of fights are presented. The number of fights and aggressors increased sharply in the Aβ group (red) compared to vehicle group (black). In the Aβ group with daily GSK2795039 injection, not a single fight was recorded during the entire hour of registration (blue). Mice actively interacted, but elements of aggression were absent. (ii) Grouping test: at the same time, direct contact (huddling) between mice was evaluated. Number of mice in a group as well as duration of grouping are presented. Mice in Aβ group (red) spent almost no time grouping, in contrast to the vehicle group (black). GSK2795039 treatment resulted in grouping dynamic improvement to or even over the vehicle levels (blue). *See Supplementary Movie 1 for sample recording of all three groups. **D** Partition Test. As direct contact with the partition correlates with the level of aggressiveness in mice, we analyzed the number of contacts (right) and time in contact with the partition (left). GSK2795039 (blue) prevented Aβ-induced aggressiveness (red) in mice when compared to vehicle (black). Data are presented as box plots with min-max whiskers and Tukey quartile method. *$p < 0.05$.

and oxidative stress[79] have both been reported to result in BACE1 upregulation and increased production of Aβ which is in turn compounded by increased Aβ release induced by the consequent network hyperactivity[80,81]. The resulting accumulation of Aβ is key to AD pathogenesis and is also known to induce oxidative stress[17].

We now show that Aβ inhibits brain glucose utilization by inducing NOX2-mediated oxidative stress, thus establishing a vicious cycle of AD pathogenesis. While the correlation between the Aβ load and NOX2 activity has been reported in multiple studies[19,28–33], until now functional consequences of this interaction were unclear. Downstream of NOX2 activation with the resulting brain hypometabolism, our results show that these changes lead to network hyperactivity and neuropsychiatric-like changes in animal behavior, establishing a causal chain between multiple AD-related pathologies.

Importantly, our data suggests that Aβ-driven disturbance of glucose metabolism is restricted mainly to cytosolic (aerobic) glycolysis while oxidative phosphorylation is actually upregulated, presumably as a compensatory mechanism as has been suggested by previous studies[37,82] (also refs. [83,38]). Our observation may seem out of line with established dogma since metabolic changes in AD brain are thought to reflect or include impaired mitochondrial function[84]. However, the underproduction of mitochondrial ATP may result from insufficient fuel provided by glycolysis as well as from impaired oxidative phosphorylation including the TCA cycle. Studies reported that early in AD, the cerebral metabolic rate of oxygen is either not altered or changes disproportionally to the prominent decrease in glucose utilization[85–87]. It was hypothesized that unchanged oxygen utilization and normal $CO_2$ production may indicate undisturbed substrate oxidation in mitochondria[85]. Moreover, other early studies that used the arterio-venous difference method showed that brain ketone uptake is still normal in moderately advanced AD[88,89], ketone catabolism being entirely mitochondrial. Studies using PET ketone tracer 11C-acetoacetate (AcAc) reported that brain metabolism of ketones is unchanged in MCI and early AD[90,91] supporting the previous assumption that oxidative phosphorylation may still be normal in early AD. Finally, a recent study utilizing RNA-seq on post-mortem AD brains showed impaired glycolytic pathways in neurons and astrocytes while the ketolytic pathways remained normal[92]. Altogether, this suggests that early brain hypometabolism in AD may be specific to glycolytic breakdown[93] and not in dysfunctional mitochondrial oxidative phosphorylation.

Our experiments suggest that NOX2 responsible for the observed Aβ effect is primarily expressed in neurons where NOX activation is mediated by NMDA receptor signaling[46], as NMDA receptor blockade by APV completely prevented the Aβ-induced glycolysis modification. We also found that a significant depletion of microglia by PLX5622 treatment had no observable effect on

Aβ toxicity, further indicating neuronal loci of NOX2 expression. However, as PLX5622 treatment did not result in complete microglial ablation and we cannot exclude the presence and role of monocyte-derived macrophages, this data remains inconclusive. Further experiments will also be needed to evaluate potential contributions of astrocytes which express both NMDARs and NOX.

The selective NOX2 antagonist GSK2795039 used in our study is a novel small-molecule NOX2 inhibitor that is a first of its kind with brain bioavailability following systematic (oral) administration. GSK2795039 is not cytotoxic at concentrations effective for NOX2 inhibition[94]: it was well tolerated in rodents, with no obvious adverse effects following 5 days of twice-daily dosing[94]. Following oral administration, GSK2795039 was detected in the blood and the central nervous system, indicating that it can cross the blood–brain barrier[94]. Thus, to our knowledge GSK2795039 is the only currently available selective NOX2 inhibitor that may be appropriate for the development of future AD therapies[95].

Given the primary role of oxidative stress in AD pathology, one other alternative treatment strategy could be scavenging ROS by antioxidants. However, clinical trials in aMCI and AD patients using antioxidants have so far been disappointing[12,96]. Our previous studies might provide an explanation for this apparent discrepancy as we showed that the potent antioxidant Tempol[97,98] failed to abate rapid ROS accumulation resulting from NOX activation during epileptic seizures[43]. It thus appears necessary to instead go to the source and to directly block pathological NOX2 activation in order to prevent its detrimental effects.

Intriguingly, oxidative stress (particularly NOX-induced oxidative stress), neuroinflammation, and hypometabolism—all acting in concert—have been reported in the early stages of other major neurodegenerative diseases[20,75,99–102]. Therefore, we posit that the role of NOX-hypometabolism-network dysfunction axis is pervasive in neurodegenerative disease initiation, and special attention should be paid to NOX2 in search of effective treatments.

One limitation of our study is the reliance on synthetic Aβ[1-42] peptide. However, previous studies from our group[53,54] and others[103] utilized a number of different synthetic and recombinant Aβ[1-42] preparations; all reported a consistent and reproducible disruptive effect on network excitability. Moreover, one major strength of our experimental protocol is the paired design with all parameters recorded both before and after addition of Aβ[1-42] in the same slice/animal, allowing us to exclude any confounding factors. Acute addition of Aβ[1-42] also excluded any secondary and chronic effects of toxicity that would be difficult to interpret. As such, we did not confirm our findings in transgenic AD mouse models; comparisons of glucose utilization in such models vs. wild-type would also be impossible outside of paired experiment paradigm, given the wide variability of glucose

transients both in vivo and in slices together with the lack of reliable normalizing factors. Nevertheless, in our previous study[53] we did find that same acute Aβ-induced modifications of glycolysis and neuronal excitability were recapitulated in ex-vivo slices from APdE9/PS1 AD model mice (see also Minkeviciene et al.[54]).

In summary, in this study we established direct causal links between $A\beta_{1-42}$, NOX2-induced oxidative stress, glucose hypometabolism, network hyperactivity, and neuropsychiatric disturbances in AD pathogenesis. We also demonstrate the potency of selective NOX2 inhibitor GSK2795039 in preventing toxic $A\beta_{1-42}$ effects and propose NOX2 as a primary target for early interventions in Alzheimer's disease, warranting further studies.

## Materials and methods

All animal protocols and experimental procedures were approved by the University of California and Gladstone Institutes under IACUC protocol AN176773, the Ethics Committees for Animal Experimentation at the INSERM (#30-03102012), and by ITEB RAS.

**Experimental animals.** Experiments were performed on mature (2–4 months) male mice. To ensure robust and reproducible results the following multiple mouse strains were utilized, with primary experimental series performed on at least two strains. In all such cases, results did not differ significantly between strains and data were subsequently pooled. OF1 (Charles River Laboratories): slice glucose/lactate, NAD(P)H/FAD and pO2 recordings; in vivo glucose recordings. C57Bl/6J mice (Jackson Labs, USA): Slice glucose, NAD(P)H/FAD and pO2 recordings, lipid peroxidation (MDA) assays. NOX2-deficient Cybb^tm1din/J mice (Jackson Labs, USA): Slice glucose, NAD(P)H/FAD/pO2 recordings, and lipid peroxidation (MDA) assays. BALB/c mice (Laboratory Animal Nursery "Pushchino", Russia): In vivo electrophysiology and glucose recordings, PLX5622 microglial depletion experiments, and behavioral experiments.

### Ex vivo experiments

*Tissue slice preparation.* Mice were anaesthetized with isoflurane and decapitated; the brain was rapidly removed from the skull and placed in the ice-cold ACSF. The ACSF solution consisted of (in mmol/L): NaCl 126, KCl 3.50, NaH2PO4 1.25, NaHCO3 25, CaCl2 2.00, MgCl2 1.30, and dextrose 5, pH 7.4. ACSF was aerated with 95% O2/5% CO2 gas mixture. Sagittal slices (350 μm) were cut using a tissue slicer (Leica VT 1200s, Leica Microsystem, Germany). During cutting, slices were submerged in an ice-cold (<6 °C) solution consisting of (in mmol/L): K-gluconate 140, HEPES 10, Na-gluconate 15, EGTA 0.2, NaCl 4, and pH adjusted to 7.2 with KOH. Slices were immediately transferred to a multi-section, dual-side perfusion holding chamber with constantly circulating ACSF and allowed to recover for 2 h at room temperature (22–24 °C).

*Synaptic stimulation and field potential recordings.* Slices were transferred to a recording chamber continuously superfused (10 ml/min) with ACSF (33–34 °C) with access to both slice sides. Schaffer collaterals/commissures were stimulated using the DS2A isolated stimulator (Digitimer Ltd, UK) with a bipolar metal electrode. LFPs were recorded using glass microelectrodes filled with ASCF, placed in stratum pyramidale of CA1 area and connected to ISO DAM-8A amplifier (WPI, FL) or MultiClamp 700B amplifier (Axon Instruments, USA). Synaptic stimulation consisted of a 30-s stimulus train (300 pulses) at 10 Hz. Stimulus current was adjusted using single pulses (40–170 μA, 200 μs) to induce a LFP with a population spike of ~50% maximal amplitude.

*Oxygen, glucose, and lactate measurements.* A Clark-style oxygen microelectrode (Unisense Ltd, Denmark) was used to measure slice tissue partial oxygen pressure (pO2). Tissue glucose and lactate concentrations were measured with enzymatic microelectrodes (tip diameter 25 μm; Sarissa Biomedical, Coventry, UK) connected to a free radical analyzer TBR4100 (Word Precision Instruments). Calibration using known multiple substrate concentrations was performed after the first polarization and was repeated following each experiment to ensure the sensor's unchanged sensitivity to a substrate.

*NAD(P)H and FAD fluorescence imaging.* Changes in NAD(P)H and FAD autofluorescence in hippocampal slices were recorded as described previously[36]. Slices were epi-illuminated using a CoolLED pE-4000 illumination system (CoolLED, USA) and imaged using a SliceScope upright microscope (Scientifica, USA) with a ×4 objective (Olympus, USA). The illumination intensity was adjusted to obtain a baseline fluorescence intensity between 2000 and 3000 optical intensity levels. NAD(P)H (excitation: 360 nm, emission: 420 nm long-pass) and FAD (excitation: 450 nm, emission: 500–570 nm) transients were elicited by network activation via 30 s repetitive synaptic stimulation as described above, and imaged in CA1 stratum pyramidale at 1 Hz. ROI intensity levels were subsequently analyzed offline using

ImageJ (NIH, USA). Data are expressed as the percentage change in fluorescence over baseline [(ΔF/F)·100].

*Pharmacology.* Antagonist of NMDA receptors, (2R)-amino-5-phosphonopentanoate (APV) was purchased from Tocris Bioscience; GSK2795039 from MedChemExpress.

*Aβ_{1−42} preparation.* Oligomeric $A\beta_{1-42}$ (Sigma Aldrich, USA) was prepared according to manufacturer specifications. Briefly, $A\beta_{1-42}$ was dissolved in 0.2% NH4OH at 1 mg/mL, brought to 400 nM concentration in ACSF and sonicated for 1 min. Prior to experiments, solution was allowed to fibrillize for minimum of 1 h.

*Lipid peroxidation assays.* Acute sagittal brain slices were prepared as described in the previous section and were allowed to recover in a vapor interface holding chamber (Scientific Systems Inc, Canada) containing standard ACSF at 34 C for 1 h. Slices were then assigned in an alternating pattern to either the control or the Aβ group, so that each slice incubated in Aβ-containing ACSF had its immediate neighbor from same hemisphere incubated in control ACSF. Following 1-h incubation in their respective oxygenated ACSF solutions, slices were snap-frozen over dry ice. Slices were then homogenized using a 26-gauge needle in the presence of a high-detergent buffer consisting of 50 mM Tris, 150 mM sodium chloride, 2% Nonidet P-40, 1% sodium deoxycholate, 4% sodium dodecyl sulfate, and supplemented with complete protease inhibitor cocktail (Roche), phosphatase inhibitor cocktail 1 (P2850, Sigma), phosphatase inhibitor cocktail 2 (P5726, Sigma), and 0.005% Butylated hydroxytoluene. The total protein in cell lysates was measured with the BCA protein assay kit (#23227, Pierce). Malondialdehyde (MDA) levels were quantified using a OxiSelect™ MDA Adduct Competitive ELISA Kit (Cell Biolabs, Inc, USA) as per manufacturer instructions. MDA levels were normalized to each sample's protein concentration and the result then normalized to the average of all control slices from same hemisphere.

### In vivo electrophysiology on freely moving animals

*Animals and surgery.* Before the experiment, mice were implanted with nichrome recording electrodes in the CA1 (AP = −2.5 mm, ML = 2 mm, H = 1.5 mm) and dentate gyrus (AP = −2mm, ML = 0.8 mm, H = 2.5 mm), and a guide cannula for i.c.v. injections (AP = −0.2 mm, ML = 1.8 mm, H = 2.2 mm). Animals were anesthetized with Zoletil (120 mg/kg) supplemented with xylazine (10 mg/kg). Electrode and cannula placements were verified post-mortem. A reference electrode was screwed into the occipital bone.

*Drug administration.* Animals received either 1 μl vehicle (NaCl 0.9% + PBS + Glucose 5 mM) or 1 μl of $A\beta_{1-42}$ (Sigma-Aldrich, 4 mg/ml in 1.0% NH4OH) intracerebroventricularly (i.c.v.) using a Hamilton syringe. In experiments with GSK2795039, 1 μl of inhibitor solution (20 mg/ml) was injected 30 min prior to $A\beta_{1-42}$ and additionally 1 μl together with $A\beta_{1-42}$ solution. After a 4-day recovery period, LFP recordings on freely moving animals were initiated. Following 60-min recordings of control activity, animals received an injection of the appropriate drug. To estimate acute effects, brain activity was monitored 90 min after the drug application. For 48 h following the experiment, animals received daily i.c.v. injection of either vehicle (control and Aβ groups) or GSK2795039 (Aβ + GSK2795039 group). 1-h long LFP recordings were performed 48 h following the acute experiments.

*LFP recordings.* LFPs were filtered (high-pass filter 0.1 Hz, low-pass filter 5 kHz) and recorded at 10 kHz sampling rate. Accumulated activity integrals and interictal spikes were detected as described previously[35,55] using custom macros in IgorPro (Wavemetrics, USA). Briefly, accumulated activity constitutes a sum of all events exceeding the 3xSD threshold and binned at 20 s intervals. Interictal spikes were detected as any event exceeding 5xSD amplitude and lasting between 10 and 90 msec. Pathological high-frequency oscillations (pHFOs) were detected as events with Hilbert transform envelope exceeding 3xSD threshold, minimum of 3 oscillations, and mean frequency between 250 and 600 Hz.

### In vivo glucose experiments on anesthetized mice

*Animals and surgery.* For acute glucose measurements, 10 mice (Aβ group, n = 5, Aβ + GSK2795039 group, n = 5) were anesthetized with pentobarbital (60 mg/kg) supplemented with xylazine (20 mg/kg). Animals were placed in a stereotaxic frame, scalped, and holes for electrodes and guide cannula were drilled. In addition to hippocampal field electrode and guide cannula, a bipolar stimulating electrode was implanted to the fimbria fornix (AP = −0.4 mm, L = 1 mm, H = 3 mm, approach angle 15°) for hippocampal network activation. A cranial window (Ø 1.5 mm) above the hippocampus was drilled ipsilateral to the cannula, and a Sarissa glucose sensor was dipped in the hippocampus (2 mm) using a micromanipulator. After surgical preparation, the cortex was kept under saline to prevent drying.

*LFPs and extracellular glucose measurements.* LFPs and extracellular glucose were measured using the same equipment as for the in vitro experiments, with network activity induced by 10 Hz, 30 s stimulation of Schaeffer collaterals and LFPs and glucose measured in CA1 region. Following 2–3 stimulations (15 min apart) in

control conditions, Aβ$_{1-42}$ or GSK2795039 followed by GSK2795039 + Aβ$_{1-42}$ were injected, with 2–3 recordings performed in each subsequent condition.

*PLX treatment for microglial depletion*. To deplete microglia, mice were switched to PLX5622 diet (PLX5622-enriched chow was provided by Research Diets, Inc., New Brunswick, USA). Mice were on the ad libitum diet for 7–15 days. Approximately 5 mg of PLX5622 was ingested daily by mice of 35 g of weight.

*Histological analysis*. Free-floating slices (35 μm) from perfused animals were incubated with Triton X (0.3%) in PBS three times for 5 min, followed by blocking solution (BSA 1%, Triton X 0.3% in PBS) for 2 h. Then, the primary antibody rabbit anti-Iba-1 (1:1000; Wako, Japan) was added and slices were incubated overnight at 4 °C. Next day, the secondary antibodies goat anti-rabbit (1:1000; Alexa Fluor 488, ThermoFisher, USA) were added for 2 h. After washout in PBS with 0.3% TritonX-100, slices were mounted on gelatinized covers in Fluoromount media (Sigma-Aldrich, USA). Immunostaining was analyzed under a Nikon E200 fluorescence microscope. In order to make a proper comparison, equivalent regions containing similar portions were chosen for all the groups. 3+ sections per animal were used for averaging. Photomicrographs using 10X (0.25 of numerical aperture) of stained fluorescence were quantified with the aid of ImageJ software (NIH, USA), and the whole hippocampus was used for quantification. The number of Iba-1+ cells was calculated using custom macros.

**Behavioral tests on effects of Aβ administration**. Male mice were housed individually with food and water ad libitum. Animals were maintained with a 12-h light/dark cycle (lights on from 9:00 a.m. to 9:00 p.m.) in temperature (22 °C ± 1 °C) -controlled room. The protocol was approved by the Committee on the Bioethics of Animal Experiments of the Institute of Theoretical and Experimental Biophysics of the Russian Academy of Sciences.

*Surgery*. Animals underwent a neurosurgical operation under general anesthesia: mixture ketamine (200 mg/kg, i.m.) and Xylazine hydrochloride (10 mg/kg, i.p.). Premedication with atropine (0.04 mg/kg) was performed subcutaneously 15 min before surgery. The body temperature was maintained by a heating pad and the cardiopulmonary parameters were monitored during the surgery by an Oxy9Vet Plus pulse oximeter (Bionet, South Korea). Guide cannula (stainless steel, 21 gauge) was implanted above the left lateral brain ventricle (AP = −0.7; L = 1.37; H = 1.25) according to the mice brain atlas[104].

*Drug administration*. Drugs were administered 1-month post-surgery to avoid surgery side effects. Mice were allocated to three groups: control (n = 6), Aβ (n = 5), and Aβ + GSK2795039 (n = 6). Injections were made i.c.v. through the guide cannula (1 μl/min) to awake mice. All substances were injected in equal volumes (1 μl) and the guide cannula was closed by a plunger. Mice in the Aβ + GSK2795039 group received GSK2795039 (MedChemExpress Europe; 4.5 mg/ml in DMSO) and oligomeric Aβ$_{1-42}$ (Sigma-Aldrich, 1 mg/ml in 1.0% NH4OH) on the first day, and daily GSK2795039 further for 14 days. Mice in the Aβ group received Aβ$_{1-42}$ on the first day and daily DMSO for the following 14 days. Mice in the control group received daily DMSO for 15 days.

*Behavioral assessment*. Experiments were carried between 18:00 and 21:00. The EthoVision system (Noldus Information Technology, Wageningen, the Netherlands) and RealTimer (Open Science, Moscow, Russia) were used for video registration and subsequent analysis.

*Open field test*. To evaluate anxiety, the animals' behavior was observed in an "open field", which was a square, black area 60 × 60 cm in size with sides 40 cm high[58]. The central area was defined as a 20 cm × 20 cm square, and the other region was defined as the peripheral area. Animals were placed initially in the center of the "field". The time spent in the field center, total travel distance and maximal velocity were recorded and analyzed.

*Social interaction test*. Social interaction tests were used on mice accustomed for 5 weeks to social isolation. To evaluate aggressive interaction in a group of animals the "fights" evaluation was used. Animals of the same experimental group were placed in the round field (diameter 50 cm) for 1 h. Each mouse was marked with a colored spot on its back for identification. Aggressive behavior estimated as number of attack defined as rushing and leaping at a partner with bites and kicks (in accordance with Mertens et al.[105]) (Supplementary Movie 1, see for example from 10:05 in Aβ group). The number of episodes of fights, the average duration of fights, the number of participants in fights, and the number of "aggressors" (mice who initiated the fight) were analyzed. Positive social interaction in groups of mice was assessed by "grouping" episodes (Supplementary Movie 1, see for example from 00:16 in control + GSK groups). Such social interaction was defined as a close and friendly grouping of 3 or more mice for at least 4 s. To estimate this type of behavior, number of grouping episodes, average duration, and number of participants were analyzed.

*Partition test*. The Partition test directly correlates with the level of aggressiveness in mice[59]. An experimental mouse was placed in a large compartment (24.1 × 17.8 × 14.0 cm) and a reference mouse in a smaller compartment (6.4 × 17.8 × 14.0 cm), with compartments separated by a see-through partition. For 5 min of the test, the number of approaches to the partition and the total time spent near the partition (when the mouse touches it with its paws or nose, reacting to a partner in the neighboring compartment) were recorded and analyzed.

## Statistics and reproducibility

Normality was determined by the Kolmogorov–Smirnov test. Significance was determined by the two-tailed Student's *t* test, Wilcoxon test, one-way ANOVA with Tukey's, or Kruskal–Wallis multiple comparisons post-hoc tests, where appropriate. Data are presented as box plots with min-max whiskers and Tukey quartile method.

**Reporting summary**. Further information on research design is available in the Nature Research Reporting Summary linked to this article.

## Data availability

All data associated with this study are available in the main text or the supplementary materials. Source data underlying figures are presented in Supplementary Data 1–2.

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

## Acknowledgements

This study was supported by the National Institute on Aging grant R01AG061150 to M.Z., RSF grants #17-75-20245 to A.M. and #20-65-46035 to I.P. and A.O. This work was also supported by NIH/NCRR grant C06 RR018928 to Gladstone Institutes. We thank Dr. Kathryn Claiborn for editorial assistance.

## Author contributions

A.M., A.I., I.P., A.O., M.Z., and S.S.J. carried out the study. M.Z. and Y.Z. designed and coordinated the study, supervised the project, analyzed data, and wrote the manuscript. S.Y.Y. managed mouse lines. Y.H. provided advice on study design and critically reviewed the manuscript.

## Competing interests

The authors declare no competing interests.
