## [Peer Review File · Communications Biology]

Reviewers' comments:

Reviewer #1 (Remarks to the Author):

COMMSBIO202719T

Malkov et al., showed that in mouse hippocampal brain slices, Abeta 1-42 application reduced glucose consumption and glycolysis, and NOX2 inhibitor GSK2795039 prevented this effect. Intracerebroventricular injection of Abeta 1-42 exerted a profound inhibitory effect on brain glucose consumption, resulting in long-lasting network hyperactivity and changes in mouse behaviors. GSK2795039 prevented all of the observed Ab-related effects. Based on these data. The authors claimed the first experimental evidence behind AD-related brain glucose hypometabolism and its consequences.

Major comments

The data presented well explain the Ab-induced glucose hypometabolism in mouse brain and involvement of NOX2-dependent ROS production, however, the reviewer concern the originality of this findings and that all conclusions depend on only inhibitors for critical molecules.

1. It has been reported that Abeta application to cultured hippocampal neurons inhibits neural glucose uptake, as listed below: The authors need to refer these publications, not only review articles cited in the current manuscript none of them refer these three articles, and carefully discuss about the consistent or inconsistent observations.

Experimental Neurology 167, 59–64 (2001)

Experimental Neurology 170, 270–276 (2001)

Experimental Neurology 174, 253–258 (2002)

2. Inhibitors used in this study are rather selective, however, they are not 100% specific for the give target molecules or not completely inhibits given targets.

(1) GSK2795039 does inhibits NOX5, so if the authors conclude that NOX2 is solely responsible for ROS produced down stream of Ab, it is essential to show the second evidence using gene targeting ot knockdown of NOX2.

(2) PLX5622 does not completely eliminate brain microglia but also affects other monocyte-derived cells in blood including macrophages. Need to show the second evidence for microglial involvement.

(3) In Figure 4B, it is difficult recognize Iba1-positive microglia, so need to replace the images with more clear images. Instead %area of Iba1 stained cells, numbers of Iba1 stained cells should be presented. Moreover, In the legend, green signals are explained as GFP fluorescence, however, their second antibody is not labeled with GFP. Please correct the legend.

(4) PMA, used for NOX2 activation, has a variety of biological effects, so it is not selective activator of NOX2

3. In this manuscript used different mouse strains for each experiment, however, it is not clear which mouse line was used in given experiment shown in each figure, and it should be explain why different mouse strains were selected for different experiments.

In page 18, "In vivo experiments on anaesthetized animals", there is no mouse strain described.

Reviewer #2 (Remarks to the Author):

This is an interesting paper that addresses questions about the potential link between Abeta accumulation and reduced glucose metabolism in AD.

Understanding the cellular mechanisms of clinically-relevant findings such as glucose hypometabolism

is extremely important, in addition the paper reveals a potential new target for therapy, which is important. Overall the paper is well written although they are a bit economical in the description of the some aspects and methods (see below). Nevertheless, there are several conceptual and technical issues that diminish my enthusiasm for the paper.

Major points:

- * The authors claim that reduced glucose uptake by Abeta drives network hyperactivity however they provide only indirect and circumstantial evidence for this link; for example the experiment illustrated in figure 1b seems to show a reduction in glucose uptake but no clear change in network activity is evident in the example trace they show, and in the slice experiments shown in figure 1a they fail to measure any neuronal activity at all
- * The authors claim that Abeta effects on glucose are dependent on neuronal NOX2 based on several pharmacological experiments; but how specific are these compounds, and wouldn't it be preferable to validate results in NOX2 null animals? Such experiments may also allow to better understand the cell-type specificity of the effects (although I would not consider the latter aspects as a requirement for the current paper)
- * The authors are a bit economical with their description of experimental detail in the results section, eg what kind of Abeta1-42 is used and at what concentration? How physiological is the concentration they are using, and are there dose-dependent effects? How do they measure glucose exactly, and how do they validate this? What kind of anaesthesia is used and at what concentration? etc.
- * The experiments seem to suggest that Abeta effects on glucose are immediate, however in patients glucose hypometabolism occurs only several years to decades after Abeta starts to accumulate and mostly in brain regions that were not studied in this manuscript (such as parietal cortex or posterior cingulum) - thus, I am worried that the present findings, while very interesting, do not provide a clear explanation for the human condition. On that note, the experiments seem to rely exclusively on (presumably synthetic?) Abeta1-42 preparations and are not validated using other Abeta preparations or mouse models, which is a weakness.

Reviewer #3 (Remarks to the Author):

The manuscript by Malkov et al is focused on the unravelling the role of NOX2 activation in beta-amyloid induced alteration of glucose metabolism. The authors showed that inhibition of NOX2 with specific inhibitor reduced effect of beta-amyloid 1-42 on glucose uptake, NADH/NADPH autofluorescence and oxygen consumption in the brain slices and in vivo model. The manuscript is potentially interesting, however, the major conclusions can be misinterpreted due to methodological issues which need to be addressed.

Comments

1. Inhibition of brain glucose utilization by β A1-42 cannot be measured only by recording the level of extracellular glucose - specifically if authors claiming the affected glycolysis and need to be proved by other methods - such a measurement of glucose uptake, activity of enzymes involved in glycolysis etc.
2. Some of major conclusions based on the NAD(P)H autofluorescence measurements. However, it is not clear what these changes means. Activation of NADPH oxidase directly change NADPH level by consumption of NADPH in this enzyme and by activation of GSH production. Glucose utilisation is in also Pentose Phosphate Pathways which produce NADPH. NADH is produced in cytosol and in TCA cycle and consumed by number enzymes including complex I in mitochondria. Considering this these measurements need to be done with appropriate controls (mitochondrial activator/inhibitor, TCA cycle inhibitors, PPP inhibitor). It need to be done also to see how big the effects (scale).
3. Page 5. " β 1-42 toxicity is prevented by blockade of NOX2" No effect of β B1-42 or NOX2 inhibitor on toxicity (cell death) was shown in this study.
- 4 page 5. " significant increase in activity-driven oxygen consumption , which together with increased

NAD(P)H oxidation amplitude could indicate upregulated mitochondrial respiration to compensate for reduced glycolysis". It can be explained by many factors but not by compensation of reduced glycolysis. TCA cycle requires end product of glycolysis to be activated and produce NADH for respiration - and cannot be activated by inhibition of inhibition of this process.

5. Discussion about the role of TCA cycle in beta-amyloid pathology (Gibson GE lab) and NADPH oxidase in mitochondrial metabolism (Abramov et al., 2004) would be beneficial.

Response to reviewers.

Reviewer #1

The data presented well explain the Ab-induced glucose hypometabolism in mouse brain and involvement of NOX2-dependent ROS production, however, the reviewer concern the originality of this findings and that all conclusions depend on only inhibitors for critical molecules.

Thank you very much for your valuable comments.

Possible involvement of A β in AD-related brain glucose hypometabolism was suggested in early studies on cultured embryonic pyramidal cells where authors measured the effects of A β_{25-35} on basal uptake of radioactive (^{14}C)glucose and reported that A β inhibited glucose uptake by disrupting glucose transport¹⁻³. These data did suggest a possible link between A β and glucose hypometabolism, although such studies of dissociated cell cultures cannot be conclusive for mechanisms operative in AD. This assumption was also confirmed by the reports that oligomeric A β_{1-42} induces oxidation of glycolytic enzymes such as enolase, pyruvate kinase and glyceraldehyde-3-phosphate dehydrogenase (reviewed by⁴). In our study, while we confirm the overall observation that A β indeed impairs glucose uptake, we a) do so by direct measurements of oligomeric A β_{1-42} effects on glucose utilization induced by network activity both in brain slices and *in vivo*, b) perform additional experiments that suggest the observed A β toxicity on glucose metabolism in intact networks is not mediated by disrupted glucose transport (see Results, p.6 and Figure S1F) and is instead underlain by impaired glycolysis, and c) investigated the entire causal chain of A β toxicity *in-vivo* starting from oxidative stress and glucose hypometabolism and leading to network dysfunction and behavioral alterations. However, the principal novelty and importance of our findings is in identifying the novel and specific molecular target (NOX2) behind these detrimental effects of A β_{1-42} . To establish the critical role of NOX2, in our experiments we used a selective NOX2 antagonist GSK2795039 as the only currently available candidate for a potential medication. To further address your comments and verify the role of NOX2, we performed additional experiments and in the revised manuscript now include data on NOX2-knockout (Cybb^{tm1din/J} mice; see Figure 2F, Results p.8) confirming that the effects of GSK2795039 are reproduced in slices deficient in NOX2.

1. It has been reported that Abeta application to cultured hippocampal neurons inhibits neural glucose uptake, as listed below: The authors need to refer these publications, not only review articles cited in the current manuscript none of them refer these three articles, and carefully discuss about the consistent or inconsistent observations.

Experimental Neurology 167, 59–64 (2001)

Experimental Neurology 170, 270–276 (2001)

Experimental Neurology 174, 253–258 (2002)

Consideration of the previous studies' results and our additional experiments addressing them was added to the Results p. 6.

2. Inhibitors used in this study are rather selective, however, they are not 100% specific for the give target molecules or not completely inhibits given targets.

(1) GSK2795039 does inhibits NOX5, so if the authors conclude that NOX2 is solely responsible for ROS produced down stream of Ab, it is essential to show the second evidence using gene targeting ot knockdown of NOX2.

GSK2795039 was demonstrated to completely inhibit NOX2 with $IC_{50} < 2.88 \mu M$ ^{5,6}. Meanwhile, GSK2795039 inhibited NOX5 by a maximum 60% at concentrations $> 100 \mu M$ ⁵. However, we included data on NOX2-knockout mice (see Fig. 2F, Table1.17-20+23, Results p.8).

(2) PLX5622 does not completely eliminate brain microglia but also affects other monocyte-derived cells in blood including macrophages. Need to show the second evidence for microglial involvement.

Our experiments show that partial microglial depletion has no effect on A β toxicity on the observed parameters, suggesting a lack of microglial involvement.

(3) In Figure 4B, it is difficult recognize Iba1-positive microglia, so need to replace the images with more clear images. Instead %area of Iba1 stained cells, numbers of Iba1 stained cells should be presented. Moreover, In the legend, green signals are explained as GFP fluorescence, however, their second antibody is not labeled with GFP. Please correct the legend.

The figure has been corrected per your comments; It has been moved to Supplementary Material as Fig. S3.

(4) PMA, used for NOX2 activation, has a variety of biological effects, so it is not selective activator of NOX2.

PMA is an activator of protein kinase C and therefore cannot be selective for NOX2 (we did not claim so in the manuscript). In our study, the role of PMA in NOX2 activation was confirmed by the application of GSK2795039 which prevented the effects of PMA.

3. In this manuscript used different mouse strains for each experiment, however, it is not clear which mouse line was used in given experiment shown in each figure, and it should be explain why different mouse strains were selected for different experiments. In page 18, "In vivo experiments on anaesthetized animals", there is no mouse strain described.

This project spanned nearly a decade and included multiple collaborations; multiple mouse strains were utilized for different experimental series. We should note that while the study was done in mice, the hope is for elucidation of a general disease mechanism ultimately applicable to humans; as such, observed A β toxicity should not be dependent on the specific mouse

strain. The fact that our results are consistent across mouse strains indicates a general principle of A β action. However, we included further detail on utilized mouse strains in Methods (page 14).

Reviewer #2

Major points:

** The authors claim that reduced glucose uptake by Abeta drives network hyperactivity however they provide only indirect and circumstantial evidence for this link; for example the experiment illustrated in figure 1b seems to show a reduction in glucose uptake but no clear change in network activity is evident in the example trace they show, and in the slice experiments shown in figure 1a they fail to measure any neuronal activity at all*

It should be noted that we did not derive our conclusion of A β -induced network hyperactivity from measurements of synaptic stimulation-induced (10Hz, 30sec) network response used to trigger metabolic transients in our study (Figures 1-3). Instead, it was drawn from *in-vivo* electrophysiology experiments (see Fig.4) showing that a single i.c.v. injection of A β ₁₋₄₂ resulted in pronounced network hyperactivity starting 1 hour following the injection, and persisted up to 48 hours later as evident by increased epileptiform spike and pathological high-frequency (“fast”) ripple rates (Fig.1C,E). Further regarding Figure 1: While we previously showed that A β induced post-synaptic hyperexcitability ^{7,8}, it also inhibited synaptic function resulting in zero net change when network is triggered by repetitive synaptic stimulation ⁸. The network activity changes under A β in acute brain slices were measured for all experiments in Figure 1A,C&D and were summated in Fig.1E (& Table1.9), showing no significant change in network response, consistent with our *in-vivo* observation. We also state this in Results (page 5). Nevertheless, we performed additional analysis to confirm that none of the observed metabolic changes correlated with changes in network activity (Results page 5, Fig. S1E).

.....

** The authors claim that Abeta effects on glucose are dependent on neuronal NOX2 based on several pharmacological experiments; but how specific are these compounds, and wouldn't it be preferable to validate results in NOX2 null animals? Such experiments may also allow to better understand the cell-type specificity of the effects (although I would not consider the latter aspects as a requirement for the current paper)*

That is a very valid concern. While we used a selective NOX2 antagonist GSK2795039 to establish the role of NOX2 ^{5,6}, we agree that it is important to verify this multiple models. We therefore performed additional experiments on NOX2-deficient mice, confirming the specificity and criticality of NOX2 role in A β toxicity (no A β effect was observed in slices from these animals). We included the resulting data in the revised manuscript (see Fig.2F, Fig. S2, Table1.17-20 + 23, and text on p.8; see also ⁹).

** The authors are a bit economical with their description of experimental detail in the results section, eg what kind of Abeta1-42 is used and at what concentration? How physiological is the concentration they are using, and are there dose-dependent effects?*

How do they measure glucose exactly, and how do they validate this? What kind of anaesthesia is used and at what concentration? etc.

Total A β (A β ₁₋₄₂ + A β ₁₋₄₀) concentration in CSF of control and AD human brains is about 3nM (In AD, the ratio is shifted: A β ₁₋₄₀ is twice smaller but A β ₁₋₄₂ 1.5 times larger)¹⁰. However, oligomeric A β (A β O) are currently widely regarded as the most toxic and pathogenic form of A β . The toxic A β O species appear to be larger than 50 kDa^{11,12} and estimation of specific A β O concentration in a mixed solution is highly problematic. Mixed A β ₁₋₄₂ solutions such as the one we used are of a highly dynamic molecule composition, with molarity and number of different molecular species contained in these solutions being essentially stochastic. We utilized 400nM which is lower than most referenced studies that typically used anywhere between 1 and 10 μ M in *in-vitro* experiments; our calculated concentrations for i.c.v. injections reflect those used in other studies¹³⁻¹⁶.

Details on tissue glucose and lactate measurements are included in the Methods (page 17), in the revised text we amended it to include details on sensor calibration that also served as the validation of sensor function.

Details on anaesthesia are included in Methods: page 19 for experiments on freely moving animals (electrophysiology) and page 20 for experiments on anesthetized mice.

.....

** The experiments seem to suggest that Abeta effects on glucose are immediate, however in patients glucose hypometabolism occurs only several years to decades after Abeta starts to accumulate and mostly in brain regions that were not studied in this manuscript (such as parietal cortex or posterior cingulum) - thus, I am worried that the present findings, while very interesting, do not provide a clear explanation for the human condition.*

We are sorry to say we cannot agree with this notion. It is well established now that although the A β deposition is the earliest detected biomarker of AD pathogenesis (about 20 years before the symptomatic stage), glucose hypometabolism follows it promptly and is also one of the earliest AD biomarkers¹⁷⁻²⁰. In fact, glucose hypometabolism was shown to be an accurate predictor of MCI-AD progression²¹⁻²³. Since neuroimaging with a limited spatial resolution was commonly used in AD patients, it was highly problematic to detect metabolic changes in hippocampus. However, a recent study using a high-resolution FDG-PET and MRI reported significant hippocampal hypometabolism in early-stage AD patients²⁴.

On that note, the experiments seem to rely exclusively on (presumably synthetic?) Abeta1-42 preparations and are not validated using other Abeta preparations or mouse models, which is a weakness.

A major problem with current mouse models is that no established models for sporadic AD are currently available^{25,26}; transgenic mice only mimic the rare genetic form of early-onset AD and; such models provide no reliable information on how sporadic AD is initiated and

progresses. In our study, we focused on the direct link between A β and AD-related pathologies that is not possible to elucidate using today's transgenic models. One major strength of our study design is the ability to perform paired comparisons (before-after). Therefore, we used intracerebroventricular A β injections for our *in vivo* studies or applied A β directly to native slices, and analysed the consequences.

Regarding the A β used, we utilized synthetic A β_{1-42} (Sigma-Aldrich, USA) widely used by researchers. However, in our previous studies on A β effects on network excitability ⁷ and on energy metabolism ⁸, we utilized different synthetic A β_{1-42} preparations which worked much to the same extent. Moreover, a different recent study on A β disruption of network gamma oscillations ²⁷ used a recombinant form of the oligomeric A β_{1-42} peptide, to the same effect (it induced neuronal hyperactivity). Therefore we feel safe in our assumption that the observed effects of A β are of a general nature and are most likely not dependent on individual preparations. It would be ideal to ultimately test our model against human AD brain-derived A β_{1-42} ; unfortunately, we were unable to acquire this peptide.

Reviewer #3

Comments

1. Inhibition of brain glucose utilization by bA1-42 cannot be measured only by recording the level of extracellular glucose - specifically if authors claiming the affected glycolysis and need to be proved by other methods - such a measurement of glucose uptake, activity of enzymes involved in glycolysis etc.

Our conclusion of glycolysis inhibition by A β was drawn from our recordings of NAD(P)H fluorescence transients in parallel with tissue glucose measurements. We previously investigated in detail the waveshape of NAD(P)H fluorescence signal ²⁸ and found that NAD(P)H overshoot corresponds largely to the process of cytoplasmic glycolysis. This overshoot association with glycolysis was also confirmed by the use of the Peredox sensor which allowed to measure solely cytoplasmic NADH signaling ²⁹. Therefore, the observed A β -induced decrease in NAD(P)H overshoot most likely represents glycolysis inhibition. We further validated this hypothesis by performing additional experiments with NAD(P)H and FAD fluorescence measured in the same slice. The results have been introduced in the revised manuscript (see Figure S1F and Results, page 6).

In addition, to verify the inhibition of glycolysis by A β we also performed additional experiments with simultaneous measurements of tissue glucose and lactate concentrations during network activity (see Figure S1A,B). These revealed that the decrease in lactate release under A β correlated with the reduction in glucose uptake (Figure S1C and Results, page 4). As lactate is the final product of glycolysis, the decrease in its release indicates inhibited glycolysis. To make sure that these changes were not mediated by disrupted glucose transport, we also repeated NAD(P)H and FAD recordings following A β wash-in and subsequent doubling of ACSF glucose concentration (Figure S1F, Results p.6). Doubling of extracellular glucose had no normalizing effect on these transients, indicating that A β inhibition of glucose utilization is due to disrupted glycolysis and not altered glucose flux. Note

also that A β has been reported to oxidize several proteins involved in the glycolysis pathway, such as enolase, pyruvate kinase and glyceraldehyde-3-phosphate dehydrogenase (GAPDH) (4).

2. Some of major conclusions based on the NAD(P)H autofluorescence measurements. However, it is not clear what these changes means. Activation of NADPH oxidase directly change NADPH level by consumption of NADPH in this enzyme and by activation of GSH production. Glucose utilisation is in also Pentose Phosphate Pathways which produce NADPH. NADH is produced in cytosol and in TCA cycle and consumed by number enzymes including complex I in mitochondria. Considering this these measurements need to be done with appropriate controls (mitochondrial activator/inhibitor, TCA cycle inhibitors, PPP inhibitor). It need to be done also to see how big the effects (scale).

Registered autofluorescence represents a combination of both NADH and NADPH. Meanwhile, activation of NOX requires NADPH, and therefore can utilize a part of NADPH generated in PPP, that may potentially lead to the decrease in detected autofluorescence. However, it has been demonstrated in multiple studies both *ex vivo* and *in vivo* that despite glycolysis inhibition, oxidative stress results in stimulation of PPP pathway and an increase in the NADPH production (up to 3-5 fold; reviewed in ³⁰). In addition, animal and human studies have shown that the PPP pathway accounts for only 2–5 % of the glucose metabolism in the adult brain ^{31–37}. Therefore, a significant contribution of NADPH to the observed decrease of autofluorescence detected seems to be highly unlikely.

3. Page 5. "A β 1-42 toxicity is prevented by blockade of NOX2" No effect of aB1-42 or NOX2 inhibitor on toxicity (cell death) was shown in this study.

By definition, drug toxicity means "The degree to which a substance (a toxin or poison) can harm humans or animals". We assumed that disruption of critical cellular functions by A β was included in toxicity specification. The investigation of a critical toxic stage such as cell death was out of the frame of our study. Meanwhile, GSK2795039 has been shown to be safe and not cytotoxic at concentrations used in our study (Hirano2015); the compound was well tolerated by rodents, with no obvious adverse effects following 5 days of oral administration.

4 page 5. " significant increase in activity-driven oxygen consumption, which together with increased NAD(P)H oxidation amplitude could indicate upregulated mitochondrial respiration to compensate for reduced glycolysis". It can be explained by many factors but not by compensation of reduced glycolysis. TCA cycle requires end product of glycolysis to be activated and produce NADH for respiration - and cannot be activated by inhibition of inhibition of this process.

It has been first demonstrated by Swerdlow and colleagues (^{38,39}; see also ⁴⁰) on neuroblastoma cells that a decrease in extracellular glucose concentration as well as a partial inhibition of glycolysis lead to the enhancement of mitochondrial respiration, presumably as a compensatory mechanism. It has been confirmed later on hippocampal slices ⁴¹ where the decrease of glucose concentration in ACSF from 10 to 2.5 mM resulted in increased oxygen consumption. We further confirm this link in our study by showing that the increase in NAD(P)H "dip" amplitude was significantly and positively

correlated with parallel increase in oxygen consumption ($r = 0.61$, $p=0.015$; Fig.S1D, Results p.4). Furthermore, as mentioned previously, our simultaneous recordings of extracellular glucose and lactate transients show a correlated decrease in lactate production/release (Fig.S1A-C), confirming that aerobic glycolysis is indeed inhibited by A β .

5. Discussion about the role of TCA cycle in beta-amyloid pathology (Gibson GE lab) and NADPH oxidase in mitochondrial metabolism (Abramov et al., 2004) would be beneficial.

Impaired glucose metabolism is one of the earliest features of AD brain. These metabolic changes are thought to reflect or include impaired mitochondrial function⁴². Meanwhile, the underproduction of mitochondrial ATP may be the consequence of either insufficient fuelling provided by glycolysis and/or impaired process of oxidative phosphorylation including the TCA cycle. The early studies reported that early in AD, the cerebral metabolic rate of oxygen was not altered or was changed disproportionately to the prominent decrease in glucose utilization⁴³⁻⁴⁵. It was hypothesized that unaltered oxygen utilization and normal CO₂ production may indicate undisturbed substrate oxidation in mitochondria⁴³. Moreover, other early studies that used the arterio-venous difference method showed that brain ketone uptake is still normal in moderately advanced AD^{46,47}, while ketone catabolism is entirely mitochondrial. Recent studies using PET ketone tracer, 11C-acetoacetate (AcAc), reported that brain metabolism of ketones is unchanged in MCI and early AD^{48,49} supporting the previous assumption that oxidative phosphorylation may still be normal in early AD. This suggests that brain hypometabolism in early AD may be specific to glucose and the primary site of metabolic abnormalities is in glycolytic glucose breakdown⁵⁰ but does not include dysfunctional mitochondrial oxidative phosphorylation. Indeed, it is difficult to suggest an explanation for the undisturbed brain ketone metabolism other than that the enzymes of mitochondrial oxidative phosphorylation continue to function relatively normally, at least early in AD. This was confirmed by a recent paper⁵¹ showing that ketolytic pathway genes in neurons and astrocytes are unaltered in post mortem AD brains, while glycolytic pathway is impaired. We have amended the revised manuscript to include this discussion (Discussion, p. 14).

1. Prapong, T., Uemura, E. & Hsu, W. H. G protein and cAMP-dependent protein kinase mediate amyloid beta-peptide inhibition of neuronal glucose uptake. *Exp. Neurol.* **167**, 59–64 (2001).
2. Prapong, T. *et al.* Amyloid beta-peptide decreases neuronal glucose uptake despite causing increase in GLUT3 mRNA transcription and GLUT3 translocation to the plasma membrane. *Exp. Neurol.* **174**, 253–258 (2002).
3. Mark, R. J., Pang, Z., Geddes, J. W., Uchida, K. & Mattson, M. P. Amyloid beta-peptide impairs glucose transport in hippocampal and cortical neurons: involvement of membrane lipid peroxidation. *J. Neurosci.* **17**, 1046–1054 (1997).
4. Butterfield, D. A. & Boyd-Kimball, D. Oxidative Stress, Amyloid- β Peptide, and Altered Key

- Molecular Pathways in the Pathogenesis and Progression of Alzheimer's Disease. *J. Alzheimers. Dis.* **62**, 1345–1367 (2018).
5. Hirano, K. *et al.* Discovery of GSK2795039, a Novel Small Molecule NADPH Oxidase 2 Inhibitor. *Antioxid. Redox Signal.* **23**, 358–374 (2015).
 6. Augsburger, F. *et al.* Pharmacological characterization of the seven human NOX isoforms and their inhibitors. *Redox Biol* **26**, 101272 (2019).
 7. Minkeviciene, R. *et al.* Fibrillar β -amyloid-induced hyperexcitability of cortical and hippocampal neurons triggers progressive epilepsy. *J. Neurosci.* **29**, 3453–3462 (2009).
 8. Zilberter, M. *et al.* Dietary energy substrates reverse early neuronal hyperactivity in a mouse model of Alzheimer's disease. *J. Neurochem.* **125**, 157–171 (2013).
 9. Park, L. *et al.* Nox2-derived radicals contribute to neurovascular and behavioral dysfunction in mice overexpressing the amyloid precursor protein. *Proceedings of the National Academy of Sciences* vol. 105 1347–1352 (2008).
 10. Bergau, N., Maul, S., Rujescu, D., Simm, A. & Navarrete Santos, A. Reduction of Glycolysis Intermediate Concentrations in the Cerebrospinal Fluid of Alzheimer's Disease Patients. *Front. Neurosci.* **13**, 871 (2019).
 11. Cline, E. N., Bicca, M. A., Viola, K. L. & Klein, W. L. The Amyloid- β Oligomer Hypothesis: Beginning of the Third Decade. *J. Alzheimers. Dis.* **64**, S567–S610 (2018).
 12. Penke, B., Szűcs, M. & Bogár, F. Oligomerization and Conformational Change Turn Monomeric β -Amyloid and Tau Proteins Toxic: Their Role in Alzheimer's Pathogenesis. *Molecules* **25**, (2020).
 13. Ahmad, R. *et al.* Lupeol, a Plant-Derived Triterpenoid, Protects Mice Brains against A β -Induced Oxidative Stress and Neurodegeneration. *Biomedicines* **8**, (2020).
 14. Amin, F. U., Shah, S. A. & Kim, M. O. Vanillic acid attenuates A β 1-42-induced oxidative stress and cognitive impairment in mice. *Sci. Rep.* **7**, 40753 (2017).
 15. Morroni, F. *et al.* Protective Effects of 6-(Methylsulfinyl)hexyl Isothiocyanate on A β 1-42-Induced Cognitive Deficit, Oxidative Stress, Inflammation, and Apoptosis in Mice. *Int. J. Mol.*

- Sci.* **19**, (2018).
16. Mokarizadeh, N. *et al.* β -Lapachone attenuates cognitive impairment and neuroinflammation in beta-amyloid induced mouse model of Alzheimer's disease. *Int. Immunopharmacol.* **81**, 106300 (2020).
 17. Gordon, B. A. *et al.* Spatial patterns of neuroimaging biomarker change in individuals from families with autosomal dominant Alzheimer's disease: a longitudinal study. *Lancet Neurol.* **17**, 241–250 (2018).
 18. Lloret, A. *et al.* When Does Alzheimer's Disease Really Start? The Role of Biomarkers. *Int. J. Mol. Sci.* **20**, (2019).
 19. Chételat, G. *et al.* Amyloid-PET and 18F-FDG-PET in the diagnostic investigation of Alzheimer's disease and other dementias. *Lancet Neurol.* **19**, 951–962 (2020).
 20. Butterfield, D. A. & Halliwell, B. Oxidative stress, dysfunctional glucose metabolism and Alzheimer disease. *Nat. Rev. Neurosci.* **20**, 148–160 (2019).
 21. Caminiti, S. P. *et al.* FDG-PET and CSF biomarker accuracy in prediction of conversion to different dementias in a large multicentre MCI cohort. *Neuroimage Clin* **18**, 167–177 (2018).
 22. Drzezga, A. *et al.* Cerebral metabolic changes accompanying conversion of mild cognitive impairment into Alzheimer's disease: a PET follow-up study. *Eur. J. Nucl. Med. Mol. Imaging* **30**, 1104–1113 (2003).
 23. Mosconi, L., Pupi, A. & De Leon, M. J. Brain glucose hypometabolism and oxidative stress in preclinical Alzheimer's disease. *Ann. N. Y. Acad. Sci.* **1147**, 180–195 (2008).
 24. Choi, E.-J. *et al.* Glucose Hypometabolism in Hippocampal Subdivisions in Alzheimer's Disease: A Pilot Study Using High-Resolution 18 F-FDG PET and 7.0-T MRI. *J. Clin. Neurol.* **14**, 158 (2018).
 25. Foidl, B. M. & Humpel, C. Can mouse models mimic sporadic Alzheimer's disease? *Neural Regeneration Res.* **15**, 401–406 (2020).
 26. Sasaguri, H. *et al.* APP mouse models for Alzheimer's disease preclinical studies. *EMBO J.* **36**, 2473–2487 (2017).

27. Kurudenkandy, F. R. *et al.* Amyloid-beta-induced action potential desynchronization and degradation of hippocampal gamma oscillations is prevented by interference with peptide conformation change and aggregation. *J. Neurosci.* **34**, 11416–11425 (2014).
28. Ivanov, A. I. *et al.* Glycolysis and oxidative phosphorylation in neurons and astrocytes during network activity in hippocampal slices. *J. Cereb. Blood Flow Metab.* **34**, 397–407 (2014).
29. Diaz-Garcia, C. M. *et al.* Neuronal Stimulation Triggers Neuronal Glycolysis and Not Lactate Uptake. *Cell Metab.* **26**, 361–374 e4 (2017).
30. Stincone, A. *et al.* The return of metabolism: biochemistry and physiology of the pentose phosphate pathway. *Biol. Rev. Camb. Philos. Soc.* **90**, 927–963 (2015).
31. Bartnik, B. L. *et al.* Upregulation of pentose phosphate pathway and preservation of tricarboxylic acid cycle flux after experimental brain injury. *J. Neurotrauma* **22**, 1052–1065 (2005).
32. Dusick, J. R. *et al.* Increased pentose phosphate pathway flux after clinical traumatic brain injury: a [1,2-¹³C₂]glucose labeling study in humans. *J. Cereb. Blood Flow Metab.* **27**, 1593–1602 (2007).
33. Hostetler, K. Y. & Landau, B. R. Estimation of the pentose cycle contribution to glucose metabolism in tissue in vivo. *Biochemistry* **6**, 2961–2964 (1967).
34. Hostetler, K. Y., Landau, B. R., White, R. J., Albin, M. S. & Yashon, D. Contribution of the pentose cycle to the metabolism of glucose in the isolated, perfused brain of the monkey. *J. Neurochem.* **17**, 33–39 (1970).
35. Gaitonde, M. K., Evison, E. & Evans, G. M. The Rate of Utilization of Glucose Via Hexosemonophosphate Shunt in Brain. *Journal of Neurochemistry* vol. 41 1253–1260 (1983).
36. Ben-Yoseph, O., Boxer, P. A. & Ross, B. D. Noninvasive assessment of the relative roles of cerebral antioxidant enzymes by quantitation of pentose phosphate pathway activity. *Neurochem. Res.* **21**, 1005–1012 (1996).
37. Morken, T. S. *et al.* Neuron–Astrocyte Interactions, Pyruvate Carboxylation and the Pentose Phosphate Pathway in the Neonatal Rat Brain. *Neurochem. Res.* **39**, 556–569 (2014).

38. Swerdlow, R. H., E, L., Aires, D. & Lu, J. Glycolysis-respiration relationships in a neuroblastoma cell line. *Biochim. Biophys. Acta* **1830**, 2891–2898 (2013).
39. Wilkins, H. M. *et al.* Oxaloacetate enhances neuronal cell bioenergetic fluxes and infrastructure. *J. Neurochem.* **137**, 76–87 (2016).
40. Zhang, T. B., Zhao, Y., Tong, Z. X. & Guan, Y. F. Inhibition of glucose-transporter 1 (GLUT-1) expression reversed Warburg effect in gastric cancer cell MKN45. *Int. J. Clin. Exp. Med.* **8**, 2423–2428 (2015).
41. Galeffi, F., Shetty, P. K., Sadgrove, M. P. & Turner, D. A. Age-related metabolic fatigue during low glucose conditions in rat hippocampus. *Neurobiol. Aging* **36**, 982–992 (2015).
42. Perez Ortiz, J. M. & Swerdlow, R. H. Mitochondrial dysfunction in Alzheimer's disease: Role in pathogenesis and novel therapeutic opportunities. *Br. J. Pharmacol.* **176**, 3489–3507 (2019).
43. Hoyer, S., Oesterreich, K. & Wagner, O. Glucose metabolism as the site of the primary abnormality in early-onset dementia of Alzheimer type? *J. Neurol.* **235**, 143–148 (1988).
44. Hoyer, S. Glucose metabolism and insulin receptor signal transduction in Alzheimer disease. *Eur. J. Pharmacol.* **490**, 115–125 (2004).
45. Hoyer, S. Oxidative energy metabolism in Alzheimer brain. Studies in early-onset and late-onset cases. *Mol. Chem. Neuropathol.* **16**, 207–224 (1992).
46. Lying-Tunell, U., Lindblad, B. S., Malmund, H. O. & Persson, B. Cerebral blood flow and metabolic rate of oxygen, glucose, lactate, pyruvate, ketone bodies and amino acids. *Acta Neurol. Scand.* **63**, 337–350 (1981).
47. Ogawa, M., Fukuyama, H., Ouchi, Y., Yamauchi, H. & Kimura, J. Altered energy metabolism in Alzheimer's disease. *J. Neurol. Sci.* **139**, 78–82 (1996).
48. Castellano, C. A. *et al.* Lower brain 18F-fluorodeoxyglucose uptake but normal 11C-acetoacetate metabolism in mild Alzheimer's disease dementia. *J. Alzheimers. Dis.* **43**, 1343–1353 (2015).
49. Croteau, E. *et al.* A cross-sectional comparison of brain glucose and ketone metabolism in cognitively healthy older adults, mild cognitive impairment and early Alzheimer's disease. *Exp.*

Gerontol. **107**, 18–26 (2018).

50. Cunnane, S. C. *et al.* Can Ketones Help Rescue Brain Fuel Supply in Later Life? Implications for Cognitive Health during Aging and the Treatment of Alzheimer's Disease. *Front. Mol. Neurosci.* **9**, 53 (2016).
51. Saito, E. R. *et al.* Alzheimer's disease alters oligodendrocytic glycolytic and ketolytic gene expression. *Alzheimers. Dement.* (2021) doi:10.1002/alz.12310.

Reviewers' comments:

Reviewer #1 (Remarks to the Author):

COMMSBIO-20-2719A-Z

The authors responded to most of my concerns appropriately, however, the authors need to provide more detailed information as shown below:

1. Figure 2F. Please provide detailed information for each line or plots, more carefully. It is impossible to identify the condition for each line or plot. For example, red lines and plots, treated with Aeta?
2. Figure S4. Please show C in the Figure. Please provide detailed information for each line or plots, more carefully. It is not clear whether the red line represents Abeta effects on PLX-treated mouse slice or not.
3. (2) PLX5622 does not completely eliminate brain microglia but also affects other monocyte-derived cells in blood including macrophages. Need to show the second evidence for microglial involvement.

The authors response

Our experiments show that partial microglial depletion has no effect on Ab toxicity on the observed parameters, suggesting a lack of microglial involvement.

This comment did not make sense at all. Partial depletion of microglia by PLX5622 resulted in a lack of microglial involvement. This experiment is not conclusive regarding microglial involvement.

4. (3). In this manuscript used different mouse strains for each experiment, however, it is not clear which mouse line was used in given experiment shown in each figure, and it should be explain why different mouse strains were selected for different experiments. In page 18, "In vivo experiments on anaesthetized animals", there is no mouse strain described.

The authors response

This project spanned nearly a decade and included multiple collaborations; multiple mouse strains were utilized for different experimental series. We should note that while the study was done in mice, the hope is for elucidation of a general disease mechanism ultimately applicable to humans; as such, observed A β toxicity should not be dependent on the specific mouse strain. The fact that our results are consistent across mouse strains indicates a general principle of A β action. However, we included further detail on utilized mouse strains in Methods (page 14).

In page 16,

Experimental animals

Experiments were performed on mature (2-4 months) male mice. A number of strains was utilized: OF1 (Charles River Laboratories, France), C57/Bl6 mice (Jackson Labs, USA), or BALB/c mice (Laboratory Animal Nursery "Pushchino", Pushchino, Russia).

Please provide exact mouse lines used in each figure. C57Bl6 is C57BL/6J or C57BL/6N or other? Please specify the subline of C57BL/6 mice used in each experiment.

Reviewer #2 (Remarks to the Author):

I thank the authors for addressing my concerns, and performing additional experiments.

However, I still feel the claim that reduced glucose uptake causes network hyperactivity is not fully supported by the data (or at least I don't see the critical experiment), and rests on circumstantial

evidence. In my opinion, this should be clarified, or tuned down.

The authors disagree with my concerns re relevance of the experiments for glucose hypometabolism in humans – I do appreciate that hypometabolism may be present in some areas earlier than previously thought but nevertheless the relationship with Abeta does not seem to be direct. For example, why do not all regions with high Abeta burden display hypometabolism? They refer to one study (Gorden et al) that finds hypometabolism in the precuneus in pre-symptomatic individuals however, importantly, not in all regions with high Abeta burden and not in the hippocampus. The other paper they refer to (Choi et al) reports hypometabolism in the hippocampus however this is all done in symptomatic AD patients in whom Abeta has of course been present for many years. I think my main point is that the relationship between Abeta and glucose hypometabolism in humans is complex and other factors probably contribute, and I feel this needs to be acknowledged

Lastly, I suggested that validating findings in mouse models or with alternative Abeta preparations would strengthen the paper, while the authors seem to think that this is not necessary. I must say that I am not convinced by their response and still feel it should be acknowledged as a limitation that they rely exclusively on synthetic Abeta 1-42 species.

Reviewer #3 (Remarks to the Author):

Authors addressed all my comments

We would like to once again thank the reviewers for their valuable input and suggestions. We have addressed their comments below.

Reviewer #1

The authors responded to most of my concerns appropriately, however, the authors need to provide more detailed information as shown below:

- 1. Figure 2F. Please provide detailed information for each line or plots, more carefully. It is impossible to identify the condition for each line or plot. For example, red lines and plots, treated with Aeta?*
- 2. Figure S4. Please show C in the Figure. Please provide detailed information for each line or plots, more carefully. It is not clear whether the red line represents Abeta effects on PLX-treated mouse slice or not.*

We agree that the figures were confusing; we modified both accordingly (added individual legends/text) to make the different conditions more distinct and clear.

- 3. (2) PLX5622 does not completely eliminate brain microglia but also affects other monocyte-derived cells in blood including macrophages. Need to show the second evidence for microglial involvement.*

The authors response

Our experiments show that partial microglial depletion has no effect on Ab toxicity on the observed parameters, suggesting a lack of microglial involvement.

This comment did not make sense at all. Partial depletion of microglia by PLX5622 resulted in a lack of microglial involvement. This experiment is not conclusive regarding microglial involvement.

The actions of PLX treatment may indeed be multi-targeted and we regret that our subsequent explanation in the initial reply was not clear. What we meant to say was that any existing and observed effect of PLX may prove to be due to factors other than microglia and would therefore require further investigation/confirmation; however, a complete lack of effect - as in our case - is unlikely to be a false negative, since we showed that depleting microglia by more than 50% has no effect on A β toxicity. We agree that these results are inconclusive indeed as we only achieved a partial microglial depletion and the remaining 50% could still be behind the observed A β toxicity. Nevertheless, when taken together with our results of NMDAR blockade (Figure S4A), they do serve to support our theory on neuronal loci of NOX2, since NMDA-activated NOX2 has been shown to be specifically expressed by neurons. We address the inconclusive nature of our PLX treatment data (Results, page 7).

- 4. (3). In this manuscript used different mouse strains for each experiment, however, it is not clear which mouse line was used in given experiment shown in each figure, and it should be explain why different mouse strains were selected for different experiments. In page 18, "In vivo experiments on anaesthetized animals", there is no mouse strain described.*

The authors response

This project spanned nearly a decade and included multiple collaborations; multiple mouse strains were utilized for different experimental series. We should note that while the study was done in mice, the hope is for elucidation of a general disease mechanism ultimately applicable to humans; as such, observed A β toxicity should not be dependent on the specific mouse strain. The fact that our results are consistent across mouse strains indicates a general principle of A β action. However, we included further detail on utilized mouse strains in Methods (page 14).

*In page 16,
Experimental animals*

Experiments were performed on mature (2-4 months) male mice. A number of strains was utilized: OF1 (Charles River Laboratories, France), C57/Bl6 mice (Jackson Labs, USA), or BALB/c mice (Laboratory Animal Nursery "Pushchino", Pushchino, Russia).

Please provide exact mouse lines used in each figure. C57Bl6 is C57BL/6J or C57BL/6N or other? Please specify the subline of C57BL/6 mice used in each experiment.

We agree that more clarity is needed for reproducibility purposes, and included the detailed specifications /discussion of choice and use of mouse strains in Methods (page 13):

Experiments were performed on mature (2-4 months) male mice. To ensure robust and reproducible results the following multiple mouse strains were utilized, with primary experimental series performed on at least two strains. In all such cases, results did not differ significantly between strains and data were subsequently pooled.

OF1 (Charles River Laboratories): Slice glucose/lactate, NAD(P)H/FAD and pO₂ recordings; in-vivo glucose recordings. C57Bl/6J mice (Jackson Labs, USA): Slice glucose, NAD(P)H/FAD and pO₂ recordings, lipid peroxidation (MDA) assays. NOX2-deficient Cybb^{tm1din}/J mice (Jackson Labs, USA): Slice glucose, NAD(P)H/FAD and pO₂ recordings, and lipid peroxidation (MDA) assays. BALB/c mice (Laboratory Animal Nursery "Pushchino", Russia): In-vivo electrophysiology and glucose recordings, PLX5622 microglial depletion experiments, and behavioral experiments.

Reviewer #2

I thank the authors for addressing my concerns, and performing additional experiments.

However, I still feel the claim that reduced glucose uptake causes network hyperactivity is not fully supported by the data (or at least I don't see the critical experiment), and rests on circumstantial evidence. In my opinion, this should be clarified, or tuned down.

We agree that the original text was not clear enough as to the full supporting evidence behind our claim of A β inducing hyperactivity via glucose hypometabolism. In this study, we observed that A β induces both glucose hypometabolism and network hyperactivity, and that blocking NOX2 activation prevented both. Although this shows a clear relationship between the two, we agree that alone this is not enough to establish causality. Actually, we addressed this very issue (outside of amyloid toxicity) in our previous study (Samokhina et al., 2017) – whether brain hypometabolism can by itself establish hyperactivity and epilepsy. We showed that chronic artificial inhibition of brain glycolysis (4 weeks of i.c.v. injections of 2-DG) resulted in epileptogenesis in initially healthy rats. We initially neglected to address this study in our manuscript, but have modified the text to include this discussion (Discussion, page 9):

Outside of A β toxicity, we have shown that chronic artificial inhibition of brain glycolysis by i.c.v. 2-deoxyglucose (2-DG) injections in healthy rats resulted in epileptogenesis and seizures (²⁵, see also ²³), proving that brain glucose hypometabolism, no matter its triggers, results in network hyperactivity.

We feel that in totality, available data allows us to make a firm conclusion that A β does indeed cause network hyperactivity by inhibiting brain glucose utilization.

The authors disagree with my concerns re relevance of the experiments for glucose hypometabolism in humans – I do appreciate that hypometabolism may be present in some areas earlier than previously thought but nevertheless the relationship with Abeta does not seem to be direct. For example, why do not all regions with

high Abeta burden display hypometabolism? They refer to one study (Gorden et al) that finds hypometabolism in the precuneus in pre-symptomatic individuals however, importantly, not in all regions with high Abeta burden and not in the hippocampus. The other paper they refer to (Choi et al) reports hypometabolism in the hippocampus however this is all done in symptomatic AD patients in whom Abeta has of course been present for many years. I think my main point is that the relationship between Abeta and glucose hypometabolism in humans is complex and other factors probably contribute, and I feel this needs to be acknowledged

We agree that the issue of A β and hypometabolism in AD patients is not so straightforward, especially at more advanced stages, and included a discussion paragraph to address available data and any possible reasons for discrepancies (Discussion, page 9):

Multiple clinical studies in MCI and early AD patients demonstrated a negative correlation between regional glucose metabolism and amyloid load^{10,62,63}. Glucose hypometabolism possesses a specific distribution pattern in the brain including precuneus and posterior cingulate cortex, extending to occipital lobes, medial and lateral frontal lobes, and middle temporal gyrus, and is consistent with the pattern of amyloid deposition^{62,63}. In addition, combined FDG-PET and CSF A β ₁₋₄₂ biomarkers have been shown to be predictive of the progression risk to AD in MCI subjects^{10,14,15}. Meanwhile, at later AD stages some studies reported poor regional correlation between hypometabolism and amyloid load^{62,64-66}, although global levels remained correlated^{10,67-70}. There may be several reasons for such discrepancy, the most likely being amyloid presence in the cortex long before the metabolic measurements, but also remote effects of amyloid deposition⁷¹ and potential differences in detection thresholds of imaging modalities, etc.⁶²

Lastly, I suggested that validating findings in mouse models or with alternative Abeta preparations would strengthen the paper, while the authors seem to think that this is not necessary. I must say that I am not convinced by their response and still feel it should be acknowledged as a limitation that they rely exclusively on synthetic Abeta 1-42 species.

We concur that the exclusive use of synthetic A β can be construed as a limitation of our study; we now address this issue directly (Discussion, page 12-13):

One limitation of our study is the reliance on synthetic A β ₁₋₄₂ peptide. However, previous studies from our group^{53,54} and others¹⁰⁰ utilized a number of different synthetic and recombinant A β ₁₋₄₂ preparations; all reported a consistent and reproducible disruptive effect on network excitability. Moreover, one major strength of our experimental protocol is the paired design with all parameters recorded both before and after addition of A β ₁₋₄₂ in the same slice/animal, allowing us to exclude any confounding factors. Acute addition of A β ₁₋₄₂ also excluded any secondary and chronic effects of toxicity that would be difficult to interpret. As such, we did not confirm our findings in transgenic AD mouse models. Comparisons of glucose utilization in such models vs. wild-type would also be impossible outside of paired experiment paradigm, given the wide variability of glucose transients both in-vivo and in slices together with the lack of reliable normalizing factors. Nevertheless, in our previous study⁵³ we did find that the acute A β -induced modifications of glycolysis and neuronal excitability were fully reproduced in ex-vivo slices from APdE9/PS1 AD model mice (see also⁵⁴).

Reviewer #3

Authors addressed all my comments

We would like to thank the Editors for their consideration and feedback on our submission. We address the specific points raised below:

After careful editorial review of your point-by-point rebuttal letter, we are still very interested in your manuscript, but do not believe that Reviewer #1's concerns about the partial microglia depletion experiments were fully addressed in the revision. Specifically, we would strongly encourage you to perform additional staining "to show the second evidence for microglial involvement," such as using TMEM119 to differentiate between resident microglia and infiltrating macrophages. At an absolute minimum, this lack of staining or resolution between cell types should be clearly mentioned as a limitation of the study in the Results and Discussion. Similarly, we believe that "the inconclusive nature of the PLX data" could also be reiterated in the Discussion (rather than just on Page 7 in the Results).

We agree that the cell specificity of NOX2 expression remains unclear given our data although the NMDAR blockade results suggest neuronal locus. We feel however that the entire subject of NOX2 expression loci is now of a supplementary status in the current manuscript and warrants a separate follow-up study. For the inconclusive nature to be more highlighted in the text we have modified the manuscript, both in the Results and Discussion section:

Results, page 7:

However, we cannot rule out a possible contribution of microglia as they have also been reported to express NMDARs⁴⁷ (although the presence of functional microglial NMDARs *in situ* is a matter of debate^{48,49}). Moreover, our microglial depletion experiments did not achieve complete ablation (Supplementary Figure 4B) and with the Iba-1 staining the potential presence of infiltrating macrophages can't be excluded.

Discussion, page 12:

Our experiments suggest that NOX2 responsible for the observed AJ3 effect is primarily expressed in neurons where NOX activation is mediated by NMDA receptor signaling⁴⁶, as NMDA receptor blockade by APV completely prevented the AJ3-induced glycolysis modification. We also found that a significant depletion of microglia by PLX5622 treatment had no observable effect on AJ3 toxicity, further indicating neuronal loci of NOX2 expression. However, as PLX5622 treatment did not result in complete microglial ablation and we also cannot exclude the presence and role of monocyte-derived macrophages, this data remains inconclusive. Further experiments will also be needed to evaluate potential contributions of astrocytes which express both NMDARs and NOX.